# Flow-of-Options: Diversified and Improved LLM Reasoning by Thinking Through Options

Lakshmi Nair [1]   Ian Trase [1]   J. Mark Kim [1]

## Abstract

We present a novel reasoning approach called Flow-of-Options (FoO), designed to address intrinsic biases in Large Language Models (LLMs). Flow-of-Options enables LLMs to systematically explore a diverse range of possibilities in their reasoning, as demonstrated by an FoO-based agentic framework developed for autonomously solving Machine Learning (ML) tasks. FoO enforces diversity in LLM solutions through compressed and interpretable task representations, resulting in improvements of 38.2% – 69.2% on standard data science tasks, and 37.4% – 47.9% on therapeutic chemistry tasks, as compared to state-of-the-art baselines. With an overall operation cost under $1 per task, our framework is well-suited for cost-sensitive applications. Going beyond tabular classification and regression, we show the broader applicability of our FoO-based agentic system to tasks such as reinforcement learning and image generation. Our code is open-sourced at: https://github.com/flagshippioneering/Flow-of-Options.

## 1. Introduction

Large Language Models (LLMs) have impacted automation on a wide spectrum of tasks, motivating a growing body of work in agentic system design, e.g., web-browsing (Yu et al., 2025), idea generation (Li et al., 2024b), and data science (Guo et al., 2024; Grosnit et al., 2024). Other approaches have focused on improving the inherent reasoning capabilities of the LLMs, through Tree-of-Thoughts (ToT) (Yao et al., 2024) or Chain-of-Thoughts (CoT) (Wei et al., 2022). This paper seeks to bridge the two areas, exploring how "thought-based" approaches can be extended to improve agentic systems. Hence, we propose ***Flow-of-Options***

[1]Flagship Pioneering, Cambridge MA, USA. Correspondence to: Lakshmi Nair <lnair@flagshippioneering.com>.

*Proceedings of the $42^{nd}$ International Conference on Machine Learning*, Vancouver, Canada. PMLR 267, 2025. Copyright 2025 by the author(s).

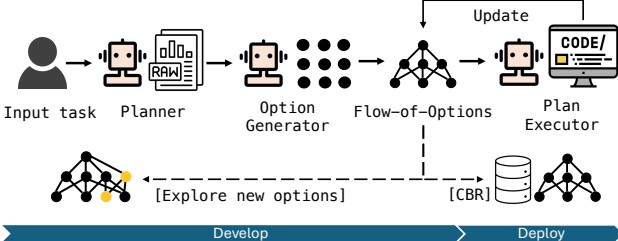

Figure 1: Overview of our Flow-of-Options based agentic framework incorporating Case-Based Reasoning (CBR).

**(FoO)**[1]. Given a task, FoO enables LLMs to "*think through*" the options available for executing each step in the task, prior to the actual execution. Broadly, FoO is a network data structure that explicitly enumerates options for each step in the task, as nodes in the network. Thus, FoO forces the LLM to be aware of, and to explore, a broader spectrum of possibilities for completing the task. We show the practical value of FoO by incorporating it in an agentic framework for automating Machine Learning (ML) tasks (Figure 1).

Recent studies show that existing agentic frameworks, such as AutoGPT (Significant Gravitas, 2023) and LangChain (Chase, 2022), often struggle with ML tasks when compared to the base LLMs (Huang et al., 2023). Alternatives involving fine-tuning LLMs (Carta et al., 2023; Christianos et al., 2023), although promising, incur significant computational effort making it challenging to adapt to broader tasks the same way that a reasoning strategy like CoT or ToT can.

Recent OpenAI models have focused on improving outputs by increasing test-time compute; fine-tuning the LLMs to "think deeper" through CoT-style reasoning (OpenAI, 2024). However, LLMs show strong biases towards options favored in their pre-training data when proposing solutions to tasks, thus impacting the agentic systems built on top of them. In Figure 2, we see that zero-shot GPT-4o and o1 have a bias towards random forest when using the Scikit-learn (Sklearn) package (Pedregosa et al., 2011), limiting the diversity of the solutions and preventing the LLMs from exploring better options for the task. Our experiments further confirm this

[1]Similar to flow-of-thought, i.e. stream of consciousness (Chafe, 2017)

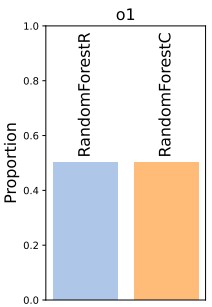 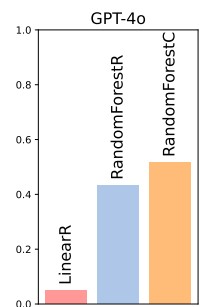 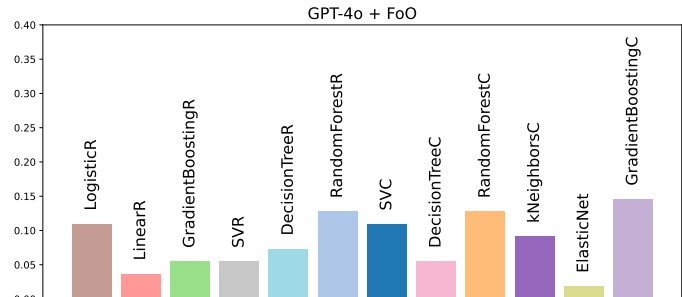

Figure 2: Proportion and diversity of LLM outputs for solving therapeutic chemistry tasks using Sklearn. FoO boosts the LLM's output diversity, enabling the LLM to explore a wider range of solutions for a task (R – regressor, C – classifier).

bias across multiple tasks. Given this pre-training bias, the model will never "*step out of its comfort zone*" without coercion, and we propose FoO as a new strategy to do this.

A recently proposed approach, DS-Agent (Guo et al., 2024) offers an alternative workaround that uses case-based reasoning (CBR) (Kolodner, 1992) to incorporate a repository of human-generated solutions within an agentic framework. CBR retrieves past solutions to problems, reuses them for the current problem, and evaluates their effectiveness, retaining the successful solutions. Through the retrieval and reuse of human insights in DS-Agent, a model is forced to explore alternatives, thus overcoming its intrinsic biases. While DS-Agent demonstrates improved performance, the system requires the use of a carefully engineered repository of human insights (expert knowledge obtained from Kaggle) summarizing the steps that worked/failed for a collection of problems. This requirement complicates the application of DS-Agent to most tasks, where a Kaggle-like, refined, large-scale repository of human insights is usually unavailable.

We find that the intrinsic knowledge of LLMs, is a sufficient proxy for human insights, *provided that the knowledge can be effectively extracted*. This motivates the value of Flow-of-Options, demonstrated through the design of an agentic framework for automated ML (Figure 3). We empirically demonstrate the benefits of our FoO-based agentic framework on 16 typical data science tasks and 20 therapeutic chemistry tasks (across ADME-Tox, drug-drug interaction, drug-target interaction, and chemical bond type prediction). ML methods designed by our approach achieves the highest overall rank compared to those designed by existing state-of-the-art baselines, with a 38.2% - 69.2% rank improvement in typical data science tasks and a 37.4% - 47.9% rank improvement in therapeutic chemistry tasks. Per-task deployment and development costs using GPT-4o are each under $1. Unlike the majority of existing work on agentic systems for data science, we further show the *broader applicability* of our work, by successfully deploying our system in domains beyond tabular classification and regression: reinforcement learning (cartpole balancing), image generation

(MNIST), and in Appendix B – clustering (unsupervised ML), machine translation, and traveling salesman problem.

## 2. Flow-of-Options

**Construction of FoO:** Given a task $\mathcal{T}$ consisting of execution steps $\mathcal{T} = \{s_1, s_2, ..., s_n\}$, we sample $k$ options for each step $i$ using an LLM $o_k^{(i)} \sim p_\theta(\mathcal{T}, s_i, o_k^{(1:i-1)})$. We prompt the LLM to emphasize diversity in its generations. By conditioning the LLM outputs on the task, the current step of the task, and the previously generated options, we ensure that the LLM outputs are consistent with the task and the options generated in previous steps of the plan. Given the LLM outputs, **Flow-of-Options** can be represented as a **network** $\mathcal{F} = (V, E, r)$ (a *directed-acyclic* graph) of depth $n$. A **node** in this network represents an option. An **edge** connects two option nodes, with associated values $r$ (initialized to a small fixed value at the beginning). To formulate the network, we first instantiate a dummy root node. Then, beginning at the root, the network is constructed with the options corresponding to each step $i$ of the plan, placed at the corresponding depth $i$. Note that LLMs are only used to generate options, and *not* to construct the FoO structure. Let $V^{(i)} = \{o_1^{(i)}, o_2^{(i)}, ..., o_k^{(i)}\}$, denote nodes at depth $i$. The edges are directed as follows:

$$E = \{(u, v) \mid u \in V^{(i-1)}, v \in V^{(i)}\} \, \forall \, i \in [1, n]$$

Where, $u$ denotes options at depth $i - 1$, $v$ denotes options at depth $i$, and $(u, v)$ represents a **directed** edge from $u$ to $v$. Visually, our FoO network can be likened to a *fully connected neural network*, where every option at depth $i - 1$ is connected to every option at depth $i$. Importantly, there are no connections between options within the same depth, i.e., $(u \in V^{(i)}, v \in V^{(i)}) \notin E$. We update the values of the edges by traversing the FoO network.

**Traversal of FoO:** Given a network $\mathcal{F}$, a **walk** $W$ represents a sequence of options, one from each depth:

$$W = (o_1^{(1)}, o_2^{(2)}, ..., o_k^{(n)}) \text{ where } o_k^{(i)} \sim V^{(i)}$$

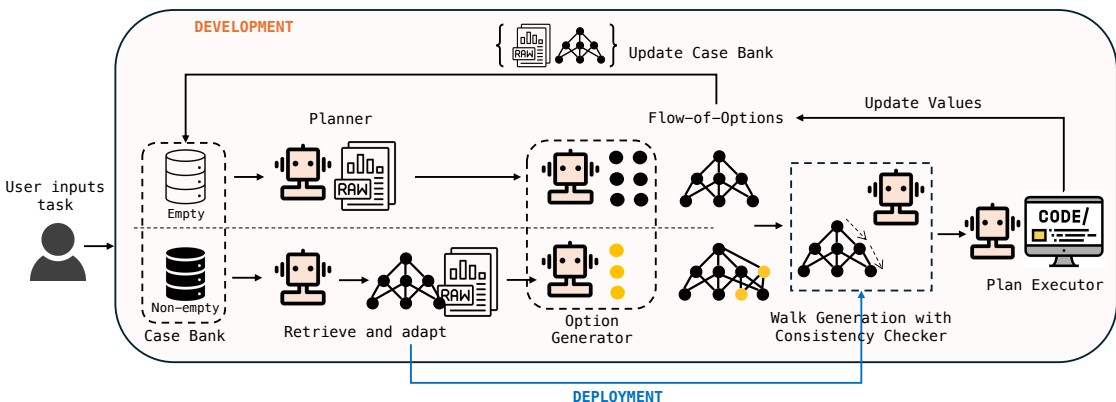

Figure 3: Overview of Flow-of-Options incorporated into an agentic framework for automating ML tasks. When starting with an empty case bank, a task plan and FoO is generated, evaluated and updated in the case bank (development). With a non-empty case bank, the closest case is retrieved and adapted to the new task. Further, either new options are explored (development), or the adapted FoO and task plan are directly used (deployment), based on the user's preference.

Our current implementation naively samples options, and some walks may be repeatedly generated from one iteration to another. In our future work, we will explore alternative informed sampling approaches. Given, a walk $W$ ($|W| = n$), we evaluate the sequence of options for the task to obtain an output value or metric $R = f(W)$. We propagate the value to the edges connecting the options using a max update:

$$r(o_k^{(i)}, o_k^{(i+1)}) = \max(r(o_k^{(i)}, o_k^{(i+1)}), R) \ \forall \ o_k^{(i)} \in W$$

Where $r(u, v)$ represents the value associated with the edge $(u, v)$. Hence, each option in $W$ will have their edge value to the subsequent option updated by the max operator. An illustrative example is shown in Figure 5.

**Beam Traversal:** Once the network is constructed and initial values are updated for the edges in $\mathcal{F}$, it would be beneficial to explore alternative combinations of the highest-valued options when generating walks. Prior work has demonstrated the value of beam search in the context of improving the output generations of language models (Graves, 2012; Franceschelli & Musolesi, 2024), by maintaining several hypotheses, eventually choosing the one with the highest probability. We follow a similar approach, and generate walks, by sampling options from the highest valued $b$ options *at every step* of the task plan. We refer to $b$ as the **beam width**. For identifying the top-$b$ options, we compute the highest value associated with each option $v \in V^{(i)}$. This is the maximum value of all the edges coming into $v$ from options in $V^{(i-1)}$:

$$value(v) = \max(r(u, v)) \ \forall \ u \in V^{(i-1)}$$

Then, the top-$b$ options in $V^{(i)}$ ($\text{Top}_b(V^{(i)})$), are the $b$ options in $V^{(i)}$ that have the highest $value$. We sample from the top-$b$ options for generating walks.

$$W_{beam} = (o_1^{(1)}, ..., o_k^{(n)}) \text{ where } o_k^{(i)} \sim \text{Top}_b(V^{(i)})$$

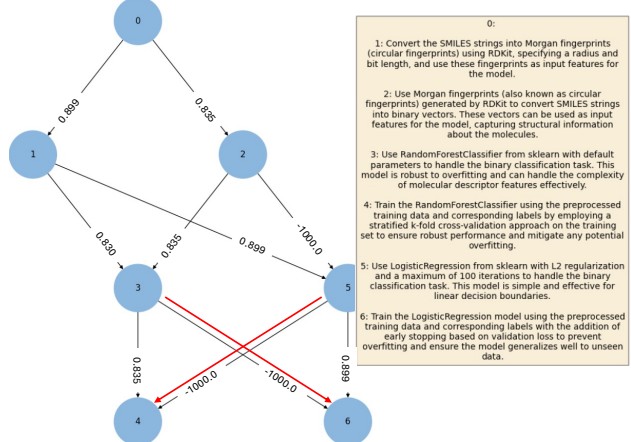

Figure 4: Example of a Flow-of-Options network ($k = 2$, $n = 3$). Red arrows denote the edges that are inconsistent as identified by the consistency checker LLM. The best walk in this case follows options $0 \rightarrow 1 \rightarrow 5 \rightarrow 6$ ($R = 0.899$).

When $b = 1$, the walk will sample the highest valued option for each step $s_i$, resulting in the best sequence of options for solving the task. Setting $b = k$ corresponds to a uniform sampling of *all* options at each step (naive). Intermediate settings explore combinations of high-valued options.

**Consistency checking:** Conceptually, the walks explore all combinations of options at different steps in the task. However, some combinations can lead to inconsistencies in the solution. For instance, if `RandomForestRegressor` is chosen at depth $i$, subsequent options in the same walk should not reference a different model. To ensure this consistency, we pass the walk $W$ through an LLM-based **consistency checker**. If the LLM identifies an inconsistent walk, it is dropped (unevaluated) and a new walk is re-sampled.

Figure 4 shows an example Flow-of-Options network. The consistency checker LLM identifies paths from $5 \to 4$ and $3 \to 6$ as inconsistent due to the discrepancy in models chosen at these steps. Next, we discuss the benefits of FoO, followed by our FoO-based agentic framework design.

## 2.1. Benefits of Flow-of-Options

Flow-of-Options is a special case of a directed acyclic graph (DAG) that is similar in structure to a fully connected neural network. *Why is this formulation useful, as opposed to a typical tree or directed acyclic graph?* We discuss this in the context of two existing methods that are closely related: SELA (Chi et al., 2024) (using trees) and Data Interpreter (Hong et al., 2024) (using DAGs). In addition to distinguishing our work from existing methods, these differences also highlight key challenges of building agentic systems for ML (Hutter et al., 2019), that we seek to alleviate with FoO.

SELA (Chi et al., 2024) decomposes tasks with pre-specified steps into a tree, combined with Monte-Carlo Tree Search (MCTS) to find optimal paths. FoO offers two improvements over this work: First, by definition, a child node in a tree can *only have one parent node*. This eliminates the possibility of exploring different combinations, e.g., of features and models. Converting Figure 4 to a tree, can cause combinations of nodes, e.g., $N2 \to N3$, to never be explored if only node $N1$ is a parent of node $N3$; Second, we argue that MCTS may be an overkill for AutoML-style problems, since it requires a significant number of rollouts for convergence, that are computationally intensive (requiring code execution). Although SELA proposes a modified upper confidence bound for trees (UCT) to mitigate this impact, SELA takes the longest among the baselines in our experiments. We believe Auto-ML can be treated differently: A "poor" rollout of a feature $F_1$ with ML model $M_1$ does not have to impact $F_1$'s value (as with UCT), if a *better* combination of $F_1$ with $M_2$ exists. If we discover *once*, that $F_1$ and $M_2$ performs well, we can stick to that path with a max update, regardless of whether $F_1$ is (on average) a "good state".

Alleviating some of the shortcomings of SELA, Data Interpreter (Hong et al., 2024) utilizes an LLM to produce a directed acyclic graph (DAG) decomposing a task into subtasks. Data Interpreter then seeks the most optimal *graph* based on an output performance measure. FoO improves over this approach in three ways: First, the LLM-generated graph in Data Interpreter *is not guaranteed to be acyclic* (as the authors note). In contrast, FoO is constructed without LLMs (only LLM-generated options). Since LLMs are not involved in the construction of the network itself, cycles are explicitly avoided. Secondly, the DAG generated by Data Interpreter, does not enforce the "fully connected" structure of FoO. As a result, Data Interpreter is not guaranteed to explore combinations of nodes at each depth, e.g., exploring

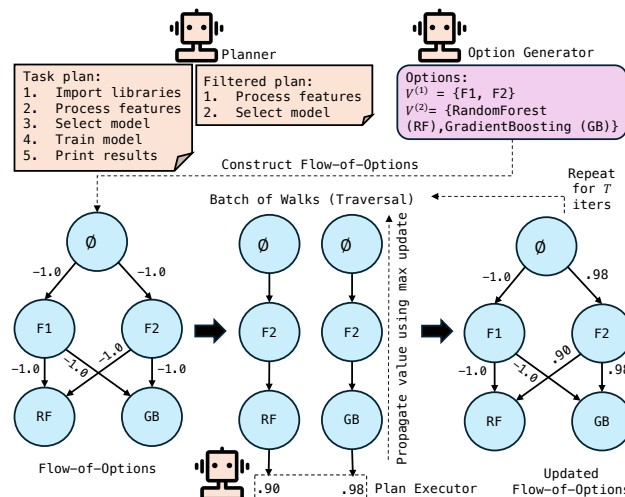

Figure 5: Illustrative example of our agentic workflow. $\Phi \to F_2$ retains the max value of $0.98$ observed on that edge.

possible combinations of feature engineering techniques with ML models, that could lead to accuracy improvements. Lastly, Data Interpreter does not enforce the exploration of different options for executing sub-tasks, often sticking to RandomForest or XGB models. In contrast, FoO enforces diversity in the options explored, resulting in the discovery of superior solutions. In summary, FoO introduces an improved, more specialized, data structure to tackle a range of ML problems. Empirically, we show that our FoO-based system yields benefits over both SELA and Data Interpreter.

## 3. Agentic Framework Using FoO

An overview of our proposed agentic framework with Flow-of-Options and Case-based Reasoning (CBR) is shown in Figure 3. We first discuss CBR and how it is incorporated with FoO before discussing the framework in more detail.

### 3.1. Case-Based Reasoning with FoO

Prior work has demonstrated the benefits of Case-Based Reasoning (CBR) in enhancing the problem-solving capabilities of LLMs, while also improving efficiency in terms of computational resources (Guo et al., 2024). Motivated by these benefits, we further incorporate CBR with Flow-of-Options, by retrieving and reusing previously generated FoO networks. CBR involves a case bank $C$ consisting of individual cases $c$. Specifically, we denote a *case* $c = (\mathcal{T}, \mathcal{F}, R^*)$, where $\mathcal{T}$ denotes the task, $\mathcal{F}$ denotes the FoO network generated for the task, and $R^*$ denotes the best reward achieved with the network for this task. Unlike the long-form descriptions of task solutions in DS-Agent (Guo et al., 2024), $\mathcal{F}$ represents *compressed* and *relevant* information on the case.

For the same task, if a new $\mathcal{F}$ with a higher reward is discov-

ered, then the case is updated with the new $\mathcal{F}$ and the corresponding reward. For a new task $\mathcal{T}'$, we retrieve the closest case $c$ that maximizes the cosine similarity of its corresponding task with $\mathcal{T}'$, i.e., $c = \arg\max_c \text{sim}(\mathbf{E}(\mathcal{T}_c), \mathbf{E}(\mathcal{T}'))$. Here, $\mathbf{E}(\cdot)$ denotes a pretrained embedding model. For the new task, we reuse the original task plan $\mathcal{T}_c$, and the corresponding $\mathcal{F}$. This ensures consistency between the retrieved Flow-of-Options and the task plan, with the implicit assumption that a similar task plan can be applied to similar tasks. We threshold the retrieval based on the similarity score, to ensure the validity of this assumption. Cases with similarity scores below a threshold will not be reused, and instead a completely new FoO will be developed for the task.

### 3.2. Framework

Inspired by prior work (Guo et al., 2024), we incorporate two phases in our framework: Development, and Deployment. Our framework begins with a user input as a string prompt, followed by either development or deployment.

**Development:** Development takes one of two paths depending on whether the case bank is empty or has data. If the case bank is empty, a task *Planner* (an LLM), generates a sequence of steps for completing the task. Given the task plan, the *Option Generator* generates options for each step of the plan. The generated options are then converted into a FoO network $\mathcal{F}$ as described in Section 2. Once the network is generated, we traverse through the network for $T$ iterations, where each iteration uses a fixed beam width that is reduced at later iterations to encourage exploration over the high-valued states (Appendix Figure 15 demonstrates the value of reducing beam width over subsequent iterations). Each iteration performs a set of $j$ walks in a batch. At the end of $j$ walks, the *Plan Executor* converts each walk into code, reflectively debugs any errors, and executes the code to extract the final metric $R_j = f(W_j)$. The metric is propagated to the nodes in the walk $W_j$ to update $\mathcal{F}$. Once $T$ iterations are complete, $\mathcal{F}$ is added into the case bank.

If the case bank is non-empty, the closest case to the user input is retrieved from the case bank. If the similarity (as described in Section 3.1) is below a pre-specified threshold, the framework reverts to the empty case bank workflow. If the similarity exceeds the threshold, the framework reuses the corresponding $\mathcal{F}$ and task plan for the current problem. First, an *Adapter* agent, adapts the $\mathcal{F}$ and corresponding task plan to the new problem, e.g., modifying regression $\rightarrow$ classification, but reusing the same class of model (`GradientBoostingRegressor` $\rightarrow$ `GradientBoostingClassifier`) (Appendix Figures 19 - 21 show additional examples of adaptations). Once adapted, the Option Generator generates new options for $\mathcal{F}$ in context of the past options in it. This enables the model to continue exploring the space with some "memory" of

its past explorations. With the updated $\mathcal{F}$, the framework proceeds with walk generations and updates values based on the corresponding code executions. The resultant $\mathcal{F}$ is then updated in the case bank. We can repeat development on the *same task*, in which case the previously generated plan and $\mathcal{F}$ for the task will be retrieved, allowing generation and exploration of new options. This can lead to the discovery of better options (improvement in accuracy), or retention of the past best performing options (stability in accuracy).

**Deployment:** The deployment phase is a computationally efficient phase with low resource requirements. Given a new task, deployment occurs when the case bank has a similar task with a high enough similarity. First, the retrieved task plan and $\mathcal{F}$ are adapted to the new task. Then deployment is analogous to development with the following setting: $k = 0$ (new options are not generated), $b = 1$, $j = 1$, and $T = 1$ ($n$ same as retrieved $\mathcal{F}$). This setting directly samples the best-valued walk, reducing deployment time and cost. The outcomes for the new task are then updated in the case bank.

**Improving Computational Efficiency:** To boost the practical applicability of our framework, particularly in the case of large FoOs, we implement three techniques. First, when the task plan is generated, we use an LLM to **filter** out the most important subset of $n$ steps that can impact accuracy on the task. Identifying this subset allows us to improve computational efficiency by restricting the depth of $\mathcal{F}$ to a few, relevant steps in the plan. Additionally, this helps narrow the exploration to the steps in the plan that *matter*, e.g., the different ways of importing a package is not critical to accuracy, and can be safely ignored. Secondly, we **prune** low-value options in $\mathcal{F}$ to prevent an explosion of the FoO size. While this risks removal of some options from "memory", we find that pruning two options per depth yields a good balance between computational efficiency and the exploration of a reasonable number of unique options. Lastly, within each iteration, we **parallelize** the $j$ walks and their corresponding code executions. This is similar to batch processing of inputs. At the end of the iteration, the values in the batch are then used to update $\mathcal{F}$.

## 4. Experiments and Results

**Baseline performance comparison on classification and regression:** We evaluate our framework on 16 tasks obtained from (Guo et al., 2024). Our baselines include DS-Agent (Guo et al., 2024), AutoGluon (Erickson et al., 2020), SELA (Chi et al., 2024), Data Interpreter (DI) (Hong et al., 2024), Autogen (Wu et al., 2024), and zero-shot with Chain-of-Thought (CoT) (Wei et al., 2022). We also evaluate on 17 ADME-Tox tasks using Therapeutic Data Commons (TDC) (Huang et al., 2021). We exclude DS-Agent from TDC tasks, since DS-Agent requires a repository of human insights (similar to Kaggle) which is currently unavailable

| | Development | | | | | | | Deployment | | | | | | | | | Avg. |
|---|---|---|---|---|---|---|---|---|---|---|---|---|---|---|---|---|---|
| | WB | MC | ES | EC | AR | ST | ILI | SS | MH | W | SD | J | CA | CS | HB | WR | Rank |
| | (↓) | (↓) | (↑) | (↑) | (↓) | (↑) | (↓) | (↑) | (↓) | (↓) | (↑) | (↓) | (↓) | (↓) | (↑) | (↑) | (↓) |
| DS-Agent | 304 | 0.30 | 0.40 | 0.27 | 4.47 | 0.78 | 6.49 | **0.99** | 0.34 | **0.34** | 0.80 | **0.67** | 0.73 | 11.7 | 0.68 | 0.65 | 3.69 |
| AutoGluon | 322 | 0.28 | 0.61 | – | – | 0.80 | – | 0.89 | 0.54 | – | 0.69 | – | 1.36 | 11.4 | – | – | 4.67 |
| SELA | 321 | 0.29 | 0.71 | – | 1.19 | 0.51 | – | 0.72 | 0.32 | – | 0.85 | 0.81 | 1.39 | 11.8 | – | 0.75 | 4.17 |
| DI | 314 | 0.30 | **0.98** | 0.43 | **1.11** | 0.82 | **1.05** | 0.88 | **0.06** | X | 0.82 | 0.98 | **0.40** | 9.88 | **0.76** | 0.75 | 2.33 |
| Autogen | 309 | 0.30 | 0.67 | 0.37 | 1.67 | 0.80 | 2.86 | 0.90 | 0.40 | 0.52 | 0.85 | 0.69 | 1.38 | 10.3 | 0.72 | **0.83** | 3.19 |
| Zero-shot | 263 | 0.26 | 0.80 | 0.35 | 1.91 | 0.78 | 5.19 | 0.83 | 0.50 | 0.38 | 0.81 | 0.79 | 1.16 | 10.0 | 0.72 | **0.83** | 3.19 |
| **Ours** | **182** | **0.18** | **0.98** | **0.43** | 1.59 | **0.82** | 1.53 | **0.99** | 0.29 | 0.36 | **0.98** | **0.67** | 0.73 | 9.18 | **0.76** | 0.80 | **1.44** |

Table 1: Performance of our framework compared to baselines on DS tasks (Task abbr. in Table 5). Results are reported over three independent runs (same prompts and fixed seed for all methods). Each run uses a different seed. Development and Deployment applies to DS-Agent and our work. Best results are in bold, with second best underlined. Systems that cannot flexibly handle non-tabular tasks are marked as –. X denotes that DI failed to generate working code in any of the three runs.

| | Development | | | | | | | | Deployment | | | | | | | | | Avg. |
|---|---|---|---|---|---|---|---|---|---|---|---|---|---|---|---|---|---|---|
| | CW | HH | BI | PG | LI | BB | PP | VD | C2 | C3 | C2S | C3S | HO | CH | A | hE | DI | Rank |
| | (↓) | (↑) | (↑) | (↑) | (↓) | (↑) | (↓) | (↑) | (↑) | (↑) | (↑) | (↑) | (↑) | (↑) | (↑) | (↑) | (↑) | (↓) |
| DeepMol | 0.35 | 0.87 | 0.50 | 0.82 | **0.69** | 0.72 | **8.34** | 0.33 | 0.20 | 0.64 | 0.38 | **0.63** | **0.49** | 0.14 | 0.65 | **0.76** | 0.79 | 2.35 |
| Autogen | 0.42 | 0.77 | 0.50 | 0.86 | 0.77 | 0.73 | 10.3 | 0.43 | 0.36 | 0.63 | 0.42 | 0.59 | 0.32 | 0.28 | **0.76** | 0.64 | 0.72 | 2.71 |
| Zero-shot | 0.44 | 0.81 | 0.56 | 0.81 | 0.76 | 0.77 | 11.2 | 0.29 | 0.34 | 0.64 | 0.39 | 0.59 | 0.28 | **0.32** | 0.71 | 0.69 | 0.72 | 2.82 |
| **Ours** | **0.34** | **0.91** | **0.58** | **0.89** | 0.75 | **0.78** | 9.51 | **0.52** | **0.57** | **0.80** | **0.62** | 0.62 | 0.26 | **0.32** | **0.76** | 0.68 | **0.84** | **1.47** |

Table 2: Performance of our FoO-based framework compared to baselines on TDC ADME-Tox tasks (Task abbr. in Table 6). Development is performed over Absorption and Distribution tasks. Deployment is performed on Metabolism, Excretion, and Toxicity tasks. Results are reported over three independent runs. Best results are in bold, with second best underlined.

for TDC. Without these insights DS-Agent performs poorly (Guo et al., 2024). Similarly, we exclude AutoGluon from TDC tasks (requiring use of packages like RDKit), and language model tasks, since AutoGluon cannot handle them. We also exclude SELA from forecasting and TDC tasks and DI from TDC tasks, as they are not flexibly supported in their MetaGPT implementations (DeepWisdom, 2024). For TDC, we compare against DeepMol (Correia et al., 2024), an AutoML approach that is specialized for ADME-Tox.

**Scaling to computationally intensive scenarios:** We evaluate our scalability via drug-drug combination, drug-target interaction prediction (TDC), and chemical bond prediction (Loschen, 2018) with $\approx 100\times$ data than the previous tasks.

**Generalizing beyond tabular classification and regression:** We evaluate our work on: a) reinforcement learning (RL) task of cartpole balancing, and b) image generation using MNIST. In the context of these tasks, we also investigate the capabilities of the *Adapter* LLM to *adapt* prior FoOs on the same task to *atypical* instantiations of the tasks.

Similar to DS-Agent, we retain a separate $\mathcal{D}_{train}$, with testing on $\mathcal{D}_{test}$. In all cases, we use GPT-4o (Hurst et al., 2024) as the foundational LLM. For Autogen, we follow the official documentation to construct a system consisting of two LLM agents: one that produces a task plan and code, and another that critiques the output to suggest improvements ("reflection") (Microsoft, 2024). We use the default param-

eterization for AutoGluon. We run SELA for 5 rollouts and DS-Agent for 5 development iterations (deployment is direct). We run DeepMol for 5 trials[2]. For development, we use $T = 5$ iterations, $j = 3$ (walks per batch), $k = 4$ (DS) or $k = 3$ (TDC), and filter tasks to use $n = 2$ (DS) or $n = 3$ (TDC). For Tables 1 and 2, we start with an empty case bank, and disable CBR for all development tasks. FoO for each development task is then added to the case bank together at the end. For deployment, we enable CBR – FoO are retrieved and reused from the case bank (no new options explored). Please refer to Appendix A for task descriptions.

### 4.1. General Data Science (DS) Tasks

We show results for 16 data science tasks in Table 1 (arrows show whether a lower (↓) or higher (↑) metric is preferred). For DS-Agent and our framework, we develop on 7/16 tasks, and deploy on the rest. We see that ML approaches designed by our framework outperforms the baselines with an average rank of 1.44 (best possible rank is 1.0, and worst is 7.0), a 38.2% to 69.2% improvement in ranks compared to baselines. All approaches, except DI, succeeded in 100% of the cases. Our approach produces high-performing solutions in 15/16 tasks (except AR). We see the accuracy benefits of FoO even in the absence of CBR, in the development phase.

---

[2]Typically run for $\approx 100$ trials for leaderboards, but we use the same number of trials for all approaches for fair comparison.

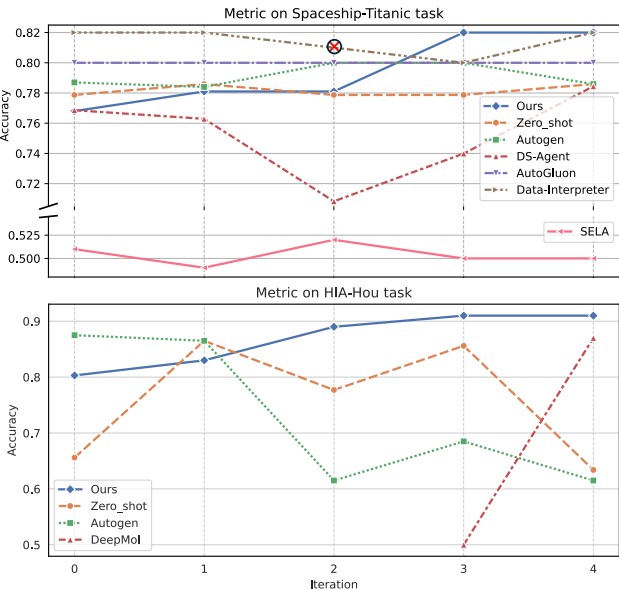

Figure 6: Our approach shows the capacity to improve. DeepMol failed in the first three iterations (returning `-inf`) and Data-Interpreter failed one iteration (Iteration 2).

| | Average Cost ($) | Average Time (mins) |
|---|---|---|
| **SELA** | 0.17 | 21.01 |
| **DS-Agent (Develop)** | 1.53 | 12.03 |
| **DS-Agent (Deploy)** | 0.06 | 0.74 |
| **Data-Interpreter** | 0.2 | 3.39 |
| **Ours (Develop)** | 0.62 | 13.29 |
| **Ours (Deploy)** | 0.03 | 0.91 |

Table 3: Resource comparisons between LLM-based agentic systems. Autogen and zero-shot consume low time and cost. Our framework has negligible deployment time and cost.

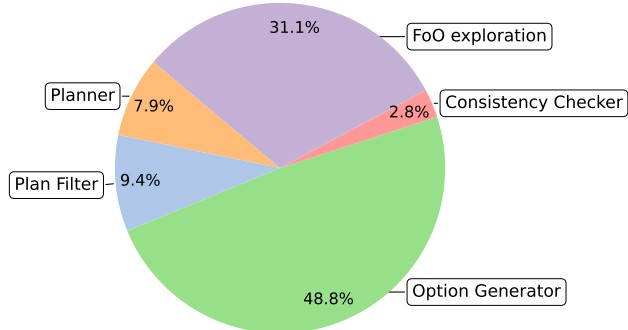

Figure 7: Ablation of total cost. Percentage of total cost contributed by each component in our FoO-based framework.

We note that Data Interpreter (DI), while competitive, exhibits two key disadvantages compared to our work: a) DI failed to produce code in 1/3 runs for 5/16 tasks; b) DI uses *highly specific, hand-crafted prompts for the tasks* (Appendix M) *compared to the general guidance provided to our FoO-based system* (Appendix N). The specificity of the prompts may, at least in part, contribute to the competitiveness of DI. Despite the more general prompts, FoO enables our approach to outperform the baselines. It is also worth noting that in spite of explicitly specifying *several* models for tabular tasks in the prompt, DI almost always used XGB or RandomForest only (See Appendix Figure 17) indicating the potential value of enumerating options in FoO form.

We demonstrate the capacity of our approach to improve on a task, by repeating development on the same task (as described in Section 3.2). Here, we run development repeatedly on the *same* task with $T = 1$ for five *separate* runs, saving the FoO at each run into the case bank. At each run, the past FoO is retrieved and *new* options are explored (we prune two options). We also run the remaining approaches for five independent runs. At each iteration $t$, DS-Agent reflects on past code from iteration $t - 1$. In Figure 6, Autogen, SELA, DI, and zero-shot performances fluctuate (since they do not reuse past experiences, leading to randomness). While DS-Agent generally improves in performance, there are cases where the agent's "reflection" results in worse outcomes (e.g., in iteration 2). Our framework reuses the past Flow-of-Options network, which includes the best solution found thus far (nodes with highest values). Exploring

new options in iterations 0 – 3, the agent finds improved outcomes. In iteration 4, it fails to find better options, achieving *stability* by retaining the best past options it has found.

Resource costs for our framework on DS tasks are in Table 3. Although costs vary, they are generally under $1. Since the repetitive LLM querying from prior work (Guo et al., 2024; Wu et al., 2024) is replaced with some non-LLM operations on FoO networks, the costs are low. Although our approach takes longer to develop, it is quick once an initial case bank is produced. SELA took the longest time compared to other baselines, possibly due to expensive MCTS rollouts. In Figure 7, we measure the percentage of total cost contributed by each component in our FoO-based framework. Majority of the total cost is attributed to the option generator and FoO exploration, which entails generating and debugging code using the LLM, for a given sequence of options (FoO walk).

### 4.2. TDC Tasks – ADME-Tox

Results are shown in Table 2. Our framework outperforms the baselines, achieving an average rank of 1.47 (37.4% to 47.9% improvement over baselines), consistently producing high-performing solutions (15/17 tasks except HO and hE).

#### 4.2.1. COMPUTATIONALLY INTENSE TASKS

Our FoO-based agentic framework involves the construction and traversal of FoO networks. In the event that the training data and models are large, it would be beneficial to scale

| Task | Zero-shot | Autogen | Ours |
|---|---|---|---|
| **DC** (TDC) ($\downarrow$) | 16.61 | 13.88 | **11.85** |
| **DTI** (TDC) ($\uparrow$) | 0.18 | 0.32 | **0.42** |
| **CB** ($\uparrow$) | 0.996 | **0.998** | **0.998** |
| **Avg. Rank** ($\downarrow$) | 2.0 | 1.3 | **1.0** |

Table 4: Performance of our FoO-based agentic system on computationally intensive therapeutic chemistry tasks (development). Results are reported on three runs.

our approach to more computationally intensive scenarios. We present one potential strategy that uses coreset selection (Guo et al., 2022) to overcome this problem. Coreset selection involves selecting a subset of the data that reasonably reflects performance on the full training data. We demonstrate the performance of this strategy on tasks with about $100\times$ more data than the previous sections. We use a simple data selection strategy that applies stratified sampling on binned label values, although more sophisticated approaches could be applied here (Guo et al., 2022). We apply development with our framework on the reduced subset of data ($\approx 1/50$th of the original dataset size), that took $\approx 15.8$ mins. Once the best performing code is generated, we apply it on the complete training set and report performances on the test set. Our results are shown in Table 4. ML approaches designed by our framework performs well across the three tasks, outperforming the baselines. Additionally, our approach explores a wider range of possibilities, as shown by the diversity of options in Figure 8, highlighting a key benefit of FoO. ML approaches designed by our framework achieves 80% - 90% of the performance of human-designed approaches in 12/19 TDC tasks (Appendix Figure 14).

### 4.3. Beyond Tabular Classification and Regression

In contrast to existing AutoML frameworks, we demonstrate our approach beyond tabular classification and regression tasks. Our baselines are Autogen and zero-shot with CoT owing to their flexibility in handling a wider range of tasks.

**Reinforcement Learning (Cartpole Balancing):** We show results of our approach on the classic cartpole balancing problem from OpenAI Gym (Brockman, 2016) in Figure 9, compared to baselines. We see a *distinct performance difference between our approach and baselines on this task*. Our framework selects the more sophisticated REINFORCE algorithm (Zhang et al., 2021) compared to either using a simple Q-table (Autogen) or random policy (zero-shot).

We further evaluate the capacity of our framework to adapt (via the *Adapter*) to novel, uncanonical variants of the task, as inspired by prior work (Lake et al., 2017). Specifically, we added the constraint of staying within the left zone of the arena. Our prompts for this task is shown in Appendix O. Appendix Figure 21 shows how the FoO developed on

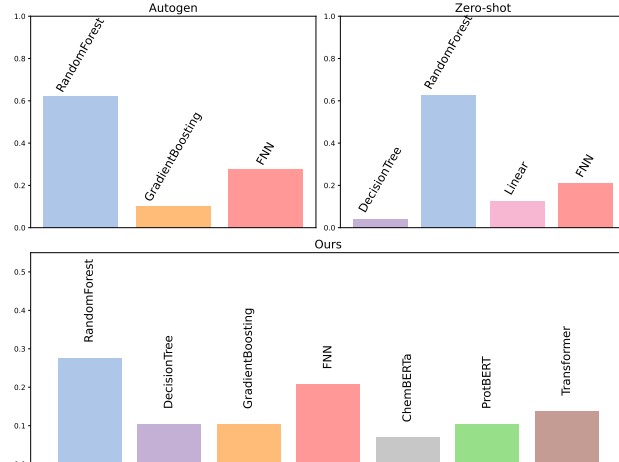

Figure 8: Proportion and diversity of ML models explored by the different AI agents across tasks from Section 4.2.1.

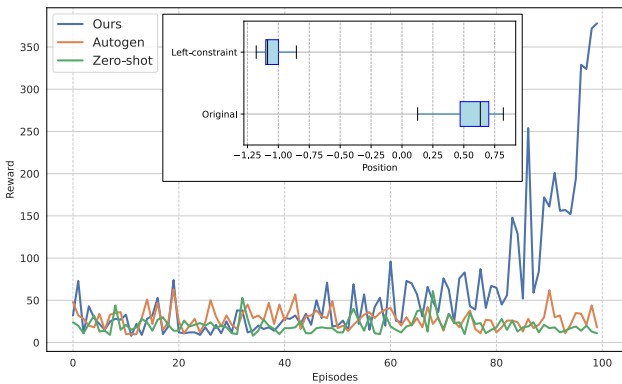

Figure 9: Reward comparisons for the different methods. **Internal:** Cartpole positions of our method for the left-constrained variant compared to the original problem.

the original problem, is successfully *adapted* with a reward modification, and deployed to the new, constrained problem. We compare the poses of the model outputs for the original and modified problems in Figure 9. We see that the adapted solution of modifying the REINFORCE reward function performs well, underscoring the capacity of our framework to effectively adapt to novel variants of past tasks.

**Synthetic Image Generation using MNIST data:** We show results of our framework on synthesis of MNIST images in Figure 10, with SSIM (Structural Similarity Index Metric) comparing our framework to the baselines. We measure SSIM between the original MNIST dataset and the generated images as a measure of similarity of the synthetic generations to the original data. Higher SSIM is preferred. We see a marked improvement in the quality of the synthetic generations for the approach developed by our framework, both quantitatively (in terms of SSIM), and qualitatively.

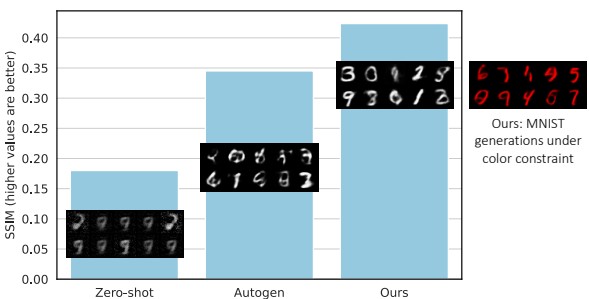

Figure 10: Output generations and SSIM values for the different methods, including output of our method for the color-constrained variant of the problem (right).

Zero-shot generation uses a Generative Adversarial Network and Autogen uses a convolutional variational-autoencoder (VAE). Our framework evaluates two different architectures for each in terms of activation functions.

We also evaluated our framework with the constraint that the generated digits should be red. The adapter proposes adding a post-processing step to the generated images to modify the channel data to generate red images (Appendix Figure 22). The results are shown to the right of Figure 10. Alongside image generation and RL, we demonstrate the application of our work to a range of additional tasks such as clustering and machine translation in Appendix B.

## 5. On the Scalability of Flow-of-Options

In section 3.2, we noted parallelization and pruning as two strategies for improving the computational efficiency of FoO. Further, we note that the consistency checker *implicitly* prunes paths that are invalid in the FoO (as shown in Figure 4). As $n$ increases, the proportion of invalid paths is also higher. Empirically, we found that the consistency checker leads to $\approx 22.8\%$ reduction in the number of paths explored. Hence, *in practice*, the computational complexity of FoO is often *lesser* than the theoretical complexity of $O(k^n)$, where $k$ is the number of options, and $n$ is the depth of the FoO. In Figure 11, we empirically measured the time taken vs. $n$ and $k$ (averaged across three runs on the CW task from Table 2). From $n = 1, k = 3$ to $n = 3, k = 3$, the total number of paths is $9\times$ higher. However, the time taken scales by $\approx 7\times$ (this includes parallelization and consistency checking, but excludes pruning which can further improve the efficiency). Moreover, the consistency checker's scaling benefit comes at a low dollar cost ($2.8\%$ of the total cost in Figure 7).

## 6. Related Work

Existing agentic designs, including recent frameworks over-fit to a narrow spectrum of tasks in data science by assuming an overall workflow (Guo et al., 2024; Wu et al., 2024; Baek

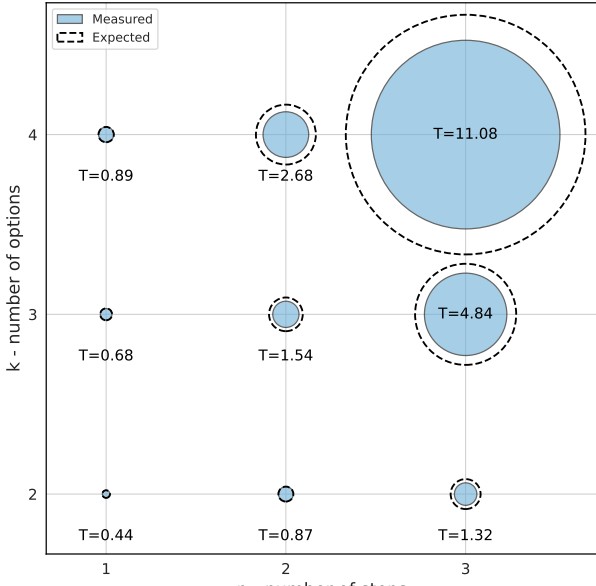

Figure 11: Scaling behavior of FoO. Measured time taken ($T$ in mins) vs. expected time for FoO exploration.

et al., 2024; Grosnit et al., 2024). Prior work has looked at Graph of Thoughts (Besta et al., 2024) to tackle reasoning problems (different from AutoML), but do not enforce the notion of "options" or option diversity. For the most closely related works, SELA (Chi et al., 2024) and Data Interpreter (Hong et al., 2024), we discussed data structure differences in Section 2.1. We highlight some additional framework-level differences here. In contrast to our work, SELA (Chi et al., 2024) predefines steps in the pipeline within the LLM prompt, e.g., data analysis, feature engineering, and model selection. Data Interpreter predefines a set of "task types" with a detailed set of instructions for each task type. This ranges from specific prompts for "feature engineering" tasks and "model training" tasks, to highly specific prompts for "image-to-web conversion" tasks. This limits their applicability to broader problems. Additionally, SELA and Data Interpreter do not leverage past experiences for new tasks.

For therapeutic tasks, prior work has introduced DrugAgent (Liu et al., 2024). Similar in spirit to our work, DrugAgent explores an "idea space" of different models but is specifically designed for the task of drug discovery only, and does not leverage structured thinking like FoO. In contrast, Flow-of-Options offers improved flexibility and performance.

## Acknowledgements

We would like to thank Mickey Atwal, Armen Mkrtchyan, and our Pioneering Intelligence teammates at Flagship Pioneering for their valuable feedback. We also thank our teammate Drew Dresser for reviewing our open-source code.

## Impact Statement

Our paper seeks to advance the field of Machine Learning. Our work is intended to *assist* (not replace) ML scientists and data scientists that use LLMs in their work, by improving the quality and value of LLM-generated outputs to user prompts. Complete autonomous usage of the system without any human supervision can pose critical security and privacy issues. Hence, precautions must be taken regarding the execution of LLM-generated code, to prevent the automatic generation and execution of potentially malicious software, or the accidental transmission of sensitive data to external servers in the case of API usage. We executed our methods within environments that have the necessary safeguards in place. The use of LLMs that adhere to strict safety and privacy regulations is heavily emphasized.

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

# Appendix Contents

# A. Task descriptions and hyperparameters

| Dataset | Abbr. | Task | Metric |
|---|---|---|---|
| Wild-blueberry-yield | WB | Regression | MAE |
| Media-campaign-cost | MC | Regression | RMLSE |
| Enzyme-substrate | ES | Classification | AUROC |
| Ethanol-concentration | EC | Classification | Accuracy |
| Airline-reviews | AR | Regression | RMSE |
| Spaceship-titanic | ST | Classification | Accuracy |
| Ili | ILI | Forecasting | MSE |
| Smoker-status | SS | Classification | AUROC |
| Mohs-hardness | MH | Regression | MedAE |
| Weather | W | Forecasting | MSE |
| Software-defects | SD | Classification | AUROC |
| Jigsaw | J | Regression | RMSE |
| Crab-age | CA | Regression | MAE |
| Concrete-strength | CS | Regression | RMSE |
| Heartbeat | HB | Classification | Accuracy |
| WebMD-reviews | WR | Classification | Accuracy |

Table 5: Tasks and corresponding abbreviations for DS-Agent tasks along with their metrics.

| Dataset | Abbr. | Task | Metric |
|---|---|---|---|
| Caco2-Wang | CW | Regression | MAE |
| HIA-Hou | HH | Classification | AUROC |
| Pgp-broccatelli | PG | Classification | AUROC |
| Bioavailability | B | Classification | AUROC |
| Lipophilicity | LI | Regression | MAE |
| BBB-Martinz | BB | Classification | AUROC |
| PPBR_AZ | PP | Regression | MAE |
| VDss_Lombardo | VD | Regression | Spearman |
| CYP2D6_Veith | C2 | Classification | AUPRC |
| CYP3A4_Veith | C3 | Classification | AUPRC |
| CYP2D6 Substrate | C2S | Classification | AUPRC |
| CYP3A4 Substrate | C3S | Classification | AUPRC |
| Half Life Obach | HO | Regression | Spearman |
| Clearance Hepatocyte | CH | Regression | Spearman |
| AMES | A | Classification | AUROC |
| hERG | hE | Classification | AUROC |
| DILI | DI | Classification | AUROC |
| Drug-drug combination | DC | Regression | MAE |
| Drug-target interaction | DTI | Correlation | Pearson Corr. |
| Bond Prediction | CB | Classification | Accuracy |

Table 6: Tasks and corresponding abbreviations for TDC ADMET and additional therapeutic chemistry tasks.

| Task | T (iters) | n (FoO depth) | k (num options) | j (walks/batch) |
|---|---|---|---|---|
| RL | 1 | 1 | 4 | 4 |
| Image Generation | 1 | 1 | 4 | 4 |
| Clustering | 3 | 3 | 3 | 3 |
| Machine Translation | 3 | 3 | 3 | 3 |
| Traveling Salesman | 1 | 1 | 3 | 3 |

Table 7: Hyperparameter settings of our framework.

# B. Tasks beyond classification and regression

We investigate the application of Flow-of-Options to tasks beyond classification and regression, namely: clustering, machine translation, traveling salesman problem, and a case-study on FoO applied to a non-code generation task, i.e., a mathematical reasoning problem.

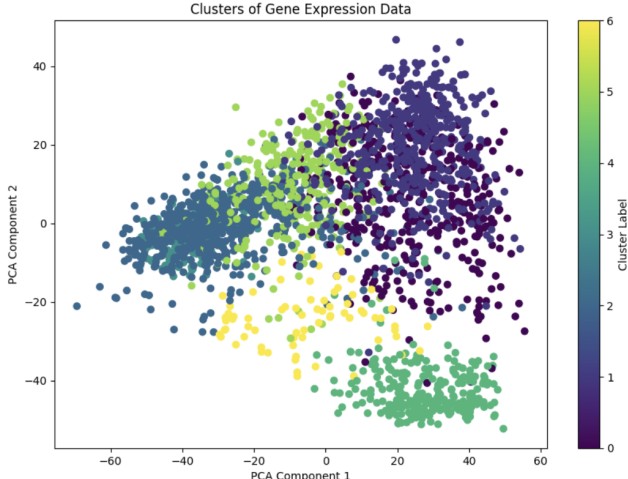

Figure 12: Visualization of clusters for the AML gene expression data generated by our system.

## B.1. Unsupervised learning – Clustering

We applied our framework to the problem of clustering gene expression data on Acute Myeloid Leukemia (AML) (Warnat-Herresthal et al., 2020). We pre-specified the number of clusters to be 7 in the input user prompt (See Appendix O). We follow an existing tutorial for pre-processing the data (Raieli, 2024) provided to our approach. The planner produces a task plan with the following filtered steps (recall that our approach filters the steps in the task plan to the most relevant steps): *1. Pre-process the data if necessary, ensuring that it is in a suitable format for clustering; 2. Select an appropriate clustering algorithm from the Sklearn library, such as KMeans, since the number of clusters is predefined; 3. Fit the clustering algorithm to the gene expression data to perform the clustering.* The resultant clustering output for the model chosen by our framework is in Figure 12. Our framework explores different options, such as scaling the data before clustering, followed by agglomerative clustering or k-means clustering. The final method chosen consists of min-max scaling of the data and k++ centroid initialization with k-means clustering to yield a Silhouette index of 0.118.

## B.2. Machine translation

We applied our framework to the problem of translating from English to French. We found that our framework ran

|  | Rouge-1 | Rouge-2 | Rouge-L |
|---|---|---|---|
| **Our approach** | 0.55 | 0.37 | 0.52 |

Table 8: Rouge scores of the ML model generated by our framework for translating from English to French.

|  | Zero-shot CoT | Autogen | Ours |
|---|---|---|---|
| Math problem | 0/3 | 0/3 | 3/3 |

Table 9: Success rate on three runs of the math problem case study. Our approach succeeds in 100% of the runs.

into issues with the download and use of Opus-100 dataset using HuggingFace API. Hence, we passed a template in our prompt to enable to model to correctly load and use the dataset API. Once provided with the API, our framework proposes the fine-tuning of various pre-trained models, such as T5 (Roberts et al., 2019), MarianMT (Junczys-Dowmunt et al., 2018), and Opus-mt (Tiedemann et al., 2022) models, available on HuggingFace. The finalized approach fine-tunes T5-base from HuggingFace on Opus-100 for English to French translation, achieving the Rouge scores reported in Table 8. We note that besides tokenization, there is no feature engineering involved in this task. Our *Planner* accounts for this specificity and generates an appropriate task plan focused on model selection, while the *Option Generator* produces appropriate options for the model choice.

### B.3. Traveling salesman problem

We applied our framework to the optimization task of solving the traveling salesman problem (TSP). For this experiment, we fixed a seed and generated a random distance matrix for the TSP with 10 cities. We note that unlike typical ML problems, there is no model selection or feature engineering involved here. Our planner produces a task plan that accounts for the nature of this task and filters the following step for option generation: *Implement a method to find the optimal path that visits each city exactly once and returns to the starting city with the minimum total distance.* Our framework explored different options for this step, including a brute-force method, genetic algorithm, and dynamic programming using Held-Karp (Shmoys & Williamson, 1990). We evaluated the paths in the FoO (the metric $R$) using *time taken to solve the task*, preferring lower metrics. Based on this metric, our framework chose the Held-Karp dynamic programming method, which completed in $\approx 0.16$ seconds, compared to $> 5$ seconds with the other approaches. This task in particular demonstrates the potential applicability of our FoO-based framework to tasks beyond data science, provided a scoring metric is available.

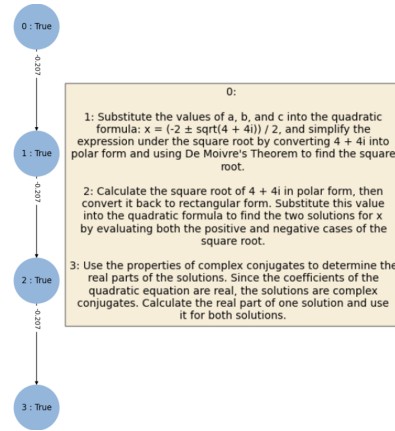

Figure 13: FoO generated for the math case study. The hyperparameter settings produce a single chain of reasoning.

### B.4. Case study with a non-code generation problem

We performed a small case study on applying our FoO-based agentic system on a math problem. The prompt provided is: "The equation $x^2 + 2x = i$ has two complex solutions. Determine the product of their real parts". This case is interesting for two reasons: 1. It doesn't necessarily involve the notion of "options" (e.g., multiple ML approaches or feature engineering), and 2. the correct evaluation metric to use (metric $R$ used to evaluate the walks is unclear). Although, existing approaches have used LLMs for scoring and evaluation (Gu et al., 2024), we circumvent the need for an evaluator for the purposes of this case study, by setting $j = 1$, $n = 3$, $k = 1$ (number of options), and $T = 1$. This setting results in a reasoning chain analogous to CoT. We find that our approach is able to produce the correct solution. The filtered task plan includes: *1. Substitute the values of a, b, and c into the quadratic formula; 2. Substitute the value of the square root back into the quadratic formula to find the two solutions for x; 3. Calculate the real parts of the two solutions obtained from the quadratic formula.* The corresponding FoO is shown in Figure 13. The model correctly produces the solution of -0.207 ($((1 - \sqrt{(2)})/2)$). We repeated this experiment for three runs by *disabling case-based reasoning*. Hence, FoOs were not saved nor retrieved. All three runs are therefore independent. We also apply Autogen and zero-shot for this task, across three independent runs. Table 9 shows the success rates of the three methods on the task, across the three runs. We note that both Autogen and zero-shot failed to produce the correct result in any of the three runs, often producing 0.0 or 1.0 as the result. We note that our approach produced "code" (the generated code is shown in J), whereas the other two approaches produced written out formulae. For instance, our approach used the `cmath` package, whereas Autogen and zero-shot explicitly described the formulae for some of

| | CW ($\downarrow$) | HH ($\uparrow$) | PG ($\uparrow$) | BI ($\uparrow$) | BB ($\uparrow$) | PP ($\downarrow$) |
|---|---|---|---|---|---|---|
| **GPT-3.5 (FoO)** | 0.43 | 0.76 | 0.85 | 0.57 | 0.69 | 10.8 |
| **GPT-3.5 (Zero-shot)** | – | 0.62 | 0.82 | 0.48 | – | – |
| **GPT-3.5 (Autogen)** | 0.43 | 0.64 | 0.82 | 0.50 | 0.61 | 11.0 |
| **GPT-4o (FoO)** | 0.34 | 0.91 | 0.89 | 0.58 | 0.78 | 9.51 |
| **GPT-4o (Zero-shot)** | 0.42 | 0.77 | 0.86 | 0.56 | 0.77 | 11.2 |
| **GPT-4o (Autogen)** | 0.44 | 0.81 | 0.81 | 0.50 | 0.73 | 10.3 |
| **Qwen2.5-Coder-32B-Instruct (FoO)** | 0.33 | 0.89 | 0.88 | 0.58 | 0.72 | 10.2 |
| **Qwen2.5-Coder-32B-Instruct (Zero-shot)** | 0.40 | 0.77 | 0.86 | – | 0.68 | 10.3 |
| **Qwen2.5-Coder-32B-Instruct (Autogen)** | 0.41 | 0.79 | 0.86 | – | 0.70 | 10.8 |
| **DeepMol** | 0.35 | 0.87 | 0.82 | 0.50 | 0.70 | 8.34 |

Table 10: Results are obtained on a subset of ADME-Tox tasks of TDC, averaged across three runs ("–" indicates that model failed to find a working solution within three runs). Qwen is an open-source LLM.

the functions that `cmath` provides, leading to errors. It is possible that the Option Generator, having been prompted to generate options (albeit one option), was more likely to think in terms of different *packages* that could be used. The code generation here loosely simulates prior approaches on "thinking through code" in LLMs (Li et al., 2024a; Chae et al., 2024), where enumerating code or pseudocode-like steps has resulted in improvements in reasoning tasks. Our future work seeks to evaluate this hypothesis more exhaustively on a range of similar problems.

## C. FoO with additional LLMs

In this section, we demonstrate the results of using GPT-3.5 and Qwen2.5-Coder-32B-Instruct, in our FoO-based agentic framework on a subset of the TDC tasks. We see in Table 10, that GPT-3.5 combined with our FoO-based agentic framework performs better than GPT-3.5 zero-shot and Autogen variants. Additionally, GPT-3.5 with FoO performs comparably to zero-shot GPT-4o. However, we note that code generation was a key challenge with GPT-3.5, leading to failures in some code executions. Most failures resulted from the use of outdated APIs (such as outdated API of RDKit) or incorrect import statements. Apart from the debugging and code generation components, GPT-3.5 was successful at generating the task plan and options, as well as consistency checking. In addition to models that are good at code generation and debugging, we note that our framework is most successful with models that are good at following instructions to produce *structured* output (e.g., output formatted as a JSON with a specific set of keys). However, with the pace of research in open-source LLMs that are competitive with closed-source LLMs like GPT-4o, we believe that these requirements will be easily satisfied by open-source LLMs in the future. Overall, FoO consistently improves performance compared to Autogen and zero-shot baselines. FoO also helps mitigate failure rates in the cases of GPT-3.5 and Qwen models.

## D. Comparison of our framework to human generated code baselines

For the TDC tasks, we compare the output of our framework to the results on the TDC leaderboard (Accessed on December 1st, 2024) in Figure 14. We note what percentage ML approaches generated by our framework achieves compared to the output metric of human-generated baseline code at #1 position on the TDC leaderboard (for ADME-Tox tasks, drug-drug, and drug-target interaction). Figure 14 shows that our approach mostly achieves $> 80\%$ of an expert human's performance. In other words, on average, it is comparable to (though not always better than) a human expert. In some cases, it outperforms the human baseline. For instance, in the drug combination (DC) task from Table 4, our approach outperforms the human baseline on the TDC leaderboard, helping democratize ML for naive users.

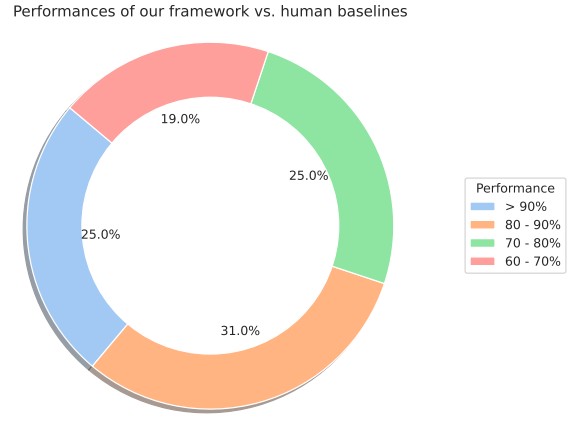

Figure 14: Comparing the accuracy of our framework to human baselines on the TDC leaderboard. The chart shows what percentages of the human baseline code metric, our framework is able to achieve.

# E. Why is beam traversal helpful?

In our experiments, we reduce the beam width in later iterations of the development phase of our framework. We find that this annealing of beam width, enables our approach to discover new combinations of the high-valued options, leading to performance improvements. We start with the full beam width $b = k$, and reduce it to $b = k/2$ at iteration 3. In Figure 15, we see that better solutions are discovered when beam width is halved across two subsequent iterations. Even though the FoO structure and options are the same, the reduced beam width encourages exploration of combinations of high-valued options (e.g., combination of a high-valued feature with a high-valued ML model), leading to discovery of improved solutions.

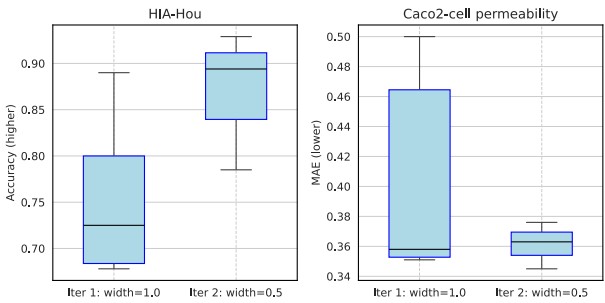

Figure 15: Improvement in metrics from iteration 1 to 2 when beam width is reduced by 50% in the second iteration.

# F. Additional component ablation study

We measured the reduction in execution time when consistency checker is added to identify invalid paths (as opposed to w/o it), and planner when the adapter is added (as opposed to w/o it). We also measure costs for each component when it is added. We measured these on average for tasks from Table 1. In Table 11, we see that both the consistency checker and plan adapter components offer benefits in terms of execution time, at minor additional cost. Without the consistency checker, the framework would explore potentially invalid paths, increasing execution costs. Similarly, by adapting the task plan (and options) from a previous task to a new task, deployment is faster. Without the adapter, a new task plan and new options would have to be explored from scratch for a new task, adding to the execution cost.

| Consistency Checker | | Planner (Adapter) | |
|---|---|---|---|
| % reduction | Cost ($) | % reduction | Cost ($) |
| 22.8 | 0.02 | 93.1 | 0.05 |

Table 11: % Reduction in execution time offered by the consistency checker and plan adapter components of our framework and their corresponding dollar costs.

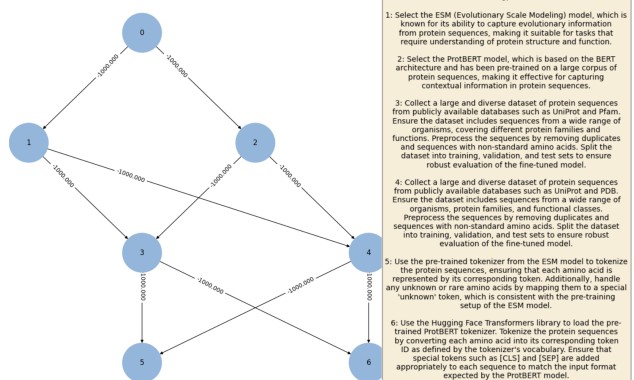

Figure 16: Example of FoO generated by incorporating external paper retrieval.

# G. Future work

In this section, we discuss two potential avenues for future work with Flow-of-Options: a) Tool-use with FoOs, and b) Nested Flow-of-Options.

### G.1. Tool use with Flow-of-Options

In our paper, FoO seeks to improve the base reasoning capabilities of the LLMs even in the absence of tools. However, tools can be a "force multiplier" with FoO. We conducted a small experiment to illustrate this further. We incorporate external research paper retrieval as a tool with FoO. The tool retrieves and provides two papers as context into the Option Generator LLM, for the example task of fine-tuning protein language models. We note the two retrieved papers incorporated into the option generator's context: ProtBERT (Elnaggar et al., 2021) and ESM (Rives et al., 2021). Figure 16 shows the resultant FoO. We see that the nodes 1 and 2 in the FoO incorporates the information in the papers. Combining external tools with Flow-of-Options can further improve the capacity of the agent to reason about tasks, and we seek to explore this synergistic potential in the future.

### G.2. Nested Flow-of-Options

Our current formulation of Flow-of-Options follows a single task plan. The task plan constructed by the planner is sufficiently high-level, that conceptually distinct options are supported via the option generator LLM. For instance, in the case of processing SMILES strings, the "feature processing" step specification in the task plan is abstract enough to accommodate RDKit-style processing and NLP-style vector embeddings as options, even though they are conceptually different. However, in some cases, a given task can be decomposed in multiple ways resulting in multiple distinct task plans that cannot be captured with a single high-level plan. In our future work, we envision a *nested* Flow-of-Options to

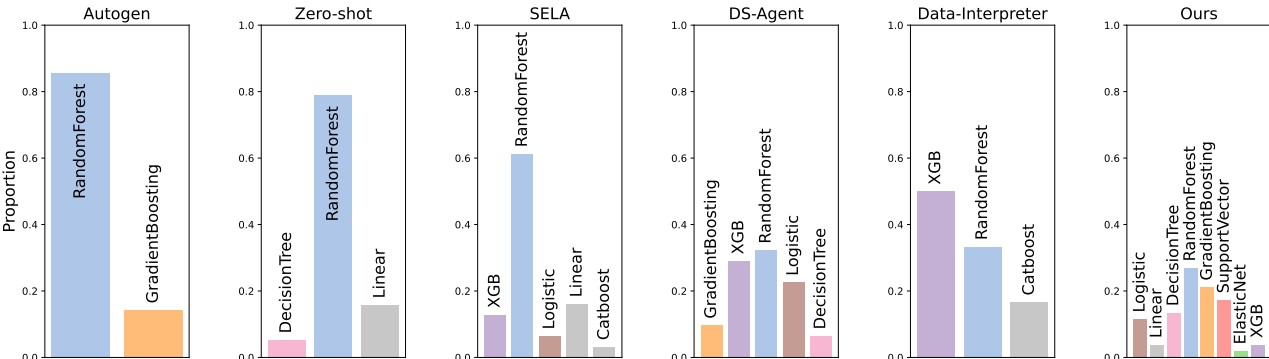

Figure 17: Diversity of explored solutions across LLM-based agents for tabular tasks from Table 1 (shown for a subset of tabular tasks – WB, MC, ES, EC, and ST – for visual clarity). Y-axis denotes proportion of explored options across the tasks.

support such cases by implementing the FoO data structure *over task plans*. In this case, the depth will be $n = 1$ and width $k$ will denote the different task decompositions. Further, each task plan "node" can be itself represented as the FoO described in this work. This would enable the agent to reason about ML *strategies*, beyond just ML methods.

## H. Relative advantages and disadvantages of the different baselines

| | Autogen | AutoML | SELA | DS-Agent | DI | Ours |
|---|---|---|---|---|---|---|
| **Metric** | 3 | 6 | 5 | 4 | 2 | 1 |
| **Diversity** | 5 | 6 | 3 | 2 | 4 | 1 |
| **Time** | 1 | 2 | 4 | 1 (Deploy) | 2 | 1 (Deploy) |
| | | | | 3 (Develop) | | 3 (Develop) |
| **Cost** | 2 | 1 | 3 | 6 | 4 | 5 |
| **Improvement** | 3 | 3 | 3 | 2 | 3 | 1 |
| **Overall Rank** | 2.8 | 3.6 | 3.6 | 3.2 | 3.0 | **2.0** |

Table 12: Ranking relative advantages and disadvantages of the methods. DI refers to Data-Interpreter.

**Quantitative:** We note the key advantages and disadvantages of the six different systems on the following dimensions: a) Metric, i.e., accuracy or error, b) Diversity – diversity of generated solutions, c) time – time taken for the system to generate a solution, d) cost – \$ cost of application of the framework, and e) Improvement – ability to improve over time to derive better solutions as the system gains more experience. Based on our results, we rank the approaches from 1 (best) – 6 (worst) on each metric (Table 12) for a summarized view of the relative advantages of the methods. AutoML refers to DeepMol and AutoGluon.

The key benefits of our approach are in terms of metric, diversity, and improvement. Our approach outperforms the baselines in terms of the output solutions (Tables 1, 2, 4), and also exhibits greater diversity in generated solutions compared to other LLM-based agents (Figures 8, 17). In terms of time (Table 3), development takes longer both with DS-Agent and our approach. However, once the system

builds an initial case bank, a user can rely on deployment alone, which is faster. We found that SELA took $\approx 21$ mins on average across the tabular tasks – the largest amount of time taken compared to the other baselines. This was attributable to the time taken by the Monte-Carlo Tree Search approach. With cost (Table 3), our framework improves on DS-Agent by replacing repeated LLM queries by operations on FoO. Lastly, our framework demonstrates the capacity for improvements over time through the incorporation of case-based reasoning with FoO (Figure 6). While DS-Agent also uses reflections to improve over past iterations, some of the reflections resulted in drops in performance across some iterations. The other LLM-based systems do not exhibit the capacity to improve over time.

**Qualitative:** We note some advantages and disadvantages of the different approaches on some qualitative measures: a) flexibility – the ability of the approach to adapt beyond a specific set of tasks, e.g., only TDC or tabular tasks, b) ease of use – ease of using the framework or approach, c) interpretability – ability for a user to understand the decision making process of the LLM.

Existing AutoML approaches overfit to tabular ML tasks by specifying the steps in the pipeline, limiting their flexibility. Data-Interpreter and SELA make underlying assumptions regarding the overall task workflows (either task types or steps in the pipeline), limiting their applicability to a broader range of tasks. Neither DS-Agent nor our approach makes similar assumptions. In terms of ease, DS-Agent requires a curated repository of human insights, unlike our framework. However, there are certain hyper-parameters to our framework that may complicate its application for some users (e.g., number of options to generate, number of steps in the task plan to filter etc.). SELA, DI, DS-Agent, and our framework offer interpretability in the thinking process of the LLMs. DS-Agent offers interpretability via the reflection logs of the LLMs showing the results of each iteration and the improvements applied in subsequent iterations based on

the previous result. SELA, DI, and our framework offers interpretability through the underlying data structures (tree, DAG, and FoO).

# I. Limitations of the overall framework

In this section, we discuss some key limitations of our work. First, our approach assumes the existence of "some" metric that can be used to evaluate the options. In the event that the metric is not clearly defined, alternative evaluators, like LLMs, can be used as a proxy for evaluating the outcomes and generating a corresponding reward (Gu et al., 2024). We also assume that the user provides some input data for each task (or it is obtained from a corresponding dataset or benchmark). If precise data is unavailable, connecting the LLM to data loading tools, can help provide the data necessary for our framework.

Second, although our approach successfully generates more diverse options for solving different tasks, we still see some residual bias for Random Forest. In the future, we seek to improve the diversity of the option generations to ensure a more uniform distribution over a range of methods. We also note that although the intrinsic knowledge of LLMs contain several methods, the LLMs are not capable of identifying more novel approaches (e.g., ChemProp for TDC tasks), that we hope to rectify through external retrieval tools.

Lastly, our approach currently naively samples walks and this results in some walks being re-sampled from one iteration to another, leading to repetition in the runs. In our future work, we seek to explore more sophisticated strategies for generating walks, such as tracking of already explored walks or updating the values to weight unexplored paths higher.

We further discuss some challenges associate with each LLM module in our framework.

## I.1. LLM-module specific challenges

While GPT-4o based frameworks were able to successfully execute tasks with 100% success rate, we note some engineering challenges for the modules in our framework:

- **Consistency Checker:** There were a few cases where the consistency checker identified an otherwise consistent pathway as inconsistent. Although the agent would simply pursue other paths, thus producing "some" working solution in each case, this mistake could cause some viable and promising paths to remain unexplored. Adding an additional "evaluator" LLM to verify the consistency response, may help mitigate this issue.

- **Retriever:** We use the same retrieval mechanism as DS-Agent (using *BAAI/LLM-embedder*). There were a few cases where the case retrieval module did not

retrieve the best case for a new task, instead retrieving an alternative case that is not as well suited to the current task. The tasks explored in our work do not seem particularly sensitive to this issue owing to the *Adapter* LLM. However, improving the retrieval module can further improve the performance of our framework.

- **Plan Executor:** Since code is generated from language descriptions of code implementations, e.g., specifying "Use `GradientBoostingClassifier`", there are slight variations from one code generation to the next, when the plan executor interprets the options. Although we found these variations to be insignificant with GPT-4o, using *code snippets* (e.g., code snippet of loading and using the model) when generating options, can potentially eliminate these differences.

- **Option Generator:** In a few cases, we noted that the option generator would produce differently phrased, but similar options in spite of being prompted to encourage diversity (See Figure 18). Introducing an additional strategy to have the LLM options be regenerated, or prompting the LLM to modify similar options may help mitigate this issue.

## J. Code generated for the math case study

**Case study: Math Problem**

```
import cmath

# Step 1: Identify the given equation: x^2 + 2x = i.
# Recognize that the equation is a quadratic equation in the form of ax^2 + bx + c
    = 0, where a = 1, b = 2, and c = -i.
a = 1
b = 2
c = -1j

# Step 2: Recall the quadratic formula for solving ax^2 + bx + c = 0, which is x =
    (-b +- sqrt(b^2 - 4ac)) / (2a).
# Implementation detail 0: Substitute the values of a, b, and c into the quadratic
    formula: x = (-2 +- sqrt(4 + 4i)) / 2
# Simplify the expression under the square root by converting 4 + 4i into polar
    form and using De Moivre's Theorem to find the square root.
delta = b**2 - 4*a*c

# Convert 4 + 4i to polar form
r = abs(delta)
theta = cmath.phase(delta)

# Calculate the square root in polar form
sqrt_r = cmath.sqrt(r)
sqrt_theta = theta / 2

# Convert back to rectangular form
sqrt_delta = sqrt_r * (cmath.cos(sqrt_theta) + 1j * cmath.sin(sqrt_theta))

# Step 3: Substitute this value into the quadratic formula to find the two
    solutions for x by evaluating both the positive and negative cases of the
    square root.
x1 = (-b + sqrt_delta) / (2*a)
x2 = (-b - sqrt_delta) / (2*a)

# Step 4: Use the properties of complex conjugates to determine the real parts of
    the solutions.
# Since the coefficients of the quadratic equation are real, the solutions are
    complex conjugates.
# Calculate the real part of one solution and use it for both solutions.
real_part_x1 = x1.real
real_part_x2 = x2.real

# Step 5: Determine the product of the real parts of the two solutions.
product_of_real_parts = real_part_x1 * real_part_x2

# Step 6: Conclude with the final result, which is the product of the real parts of
    the solutions.
print('The product of the real parts of the solutions is:', product_of_real_parts)
```

## K. Additional Flow-of-Options network examples

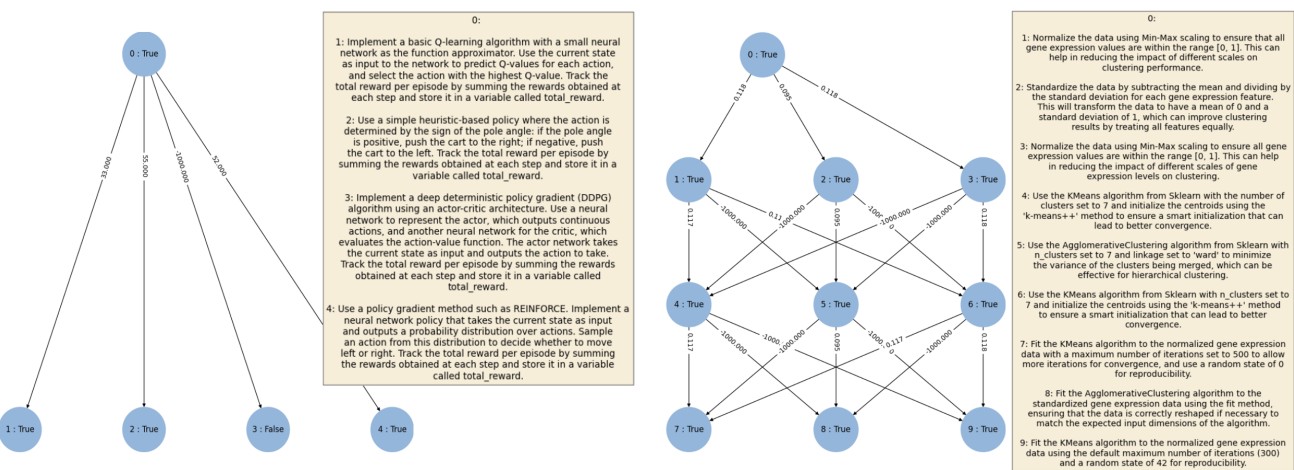

Figure 18: Examples of flow-of-options networks with varying depths (shown for the RL and clustering tasks). Paths with a value of $-1000.0$ are unexplored paths (either due to consistency check failures or because the walks were not sampled). The tree on the right demonstrates one of the system's limitations regarding generation of similar options in some cases (e.g., Nodes 1 and 3).

## L. Task adaptation using LLM adapter

In this section, we show examples of how the LLM Adapter adapts a network from one task to another under different scenarios. This is shown in Figures 19, 20, 21, and 22. The figures show the description of the nodes in the Flow-of-Options network.

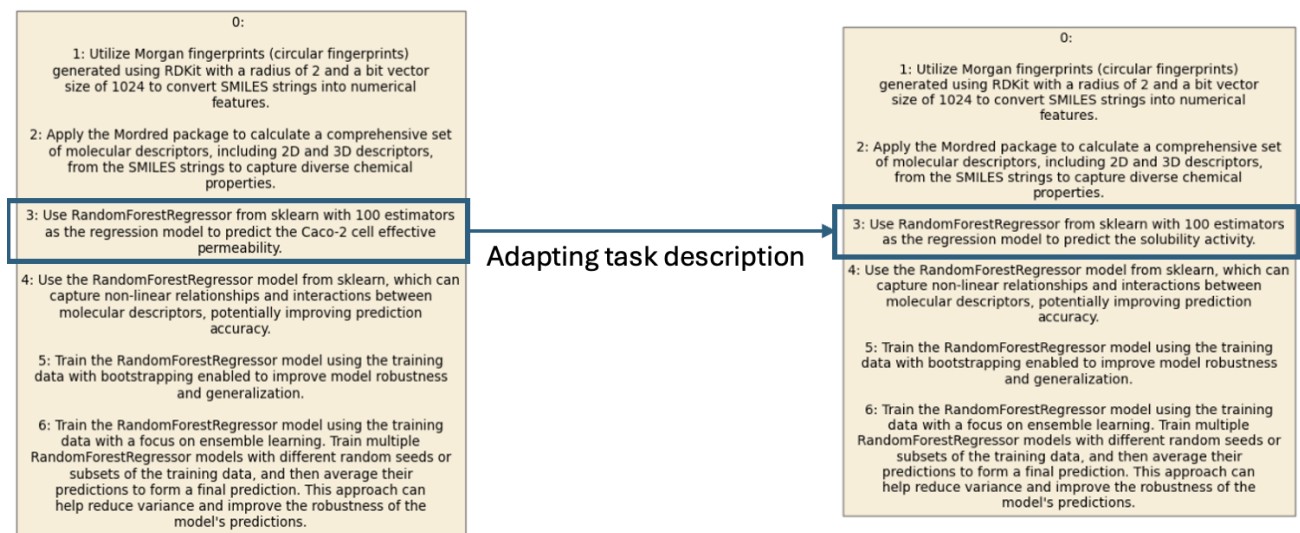

Figure 19: Example of the adapter LLM adapting the task description from Caco-2 cell permeability to Solubility prediction task in TDC. Each number marks a node in the network along with its description. The network structure is unchanged since the network is reused, but the description of the nodes are adapted to the new task.

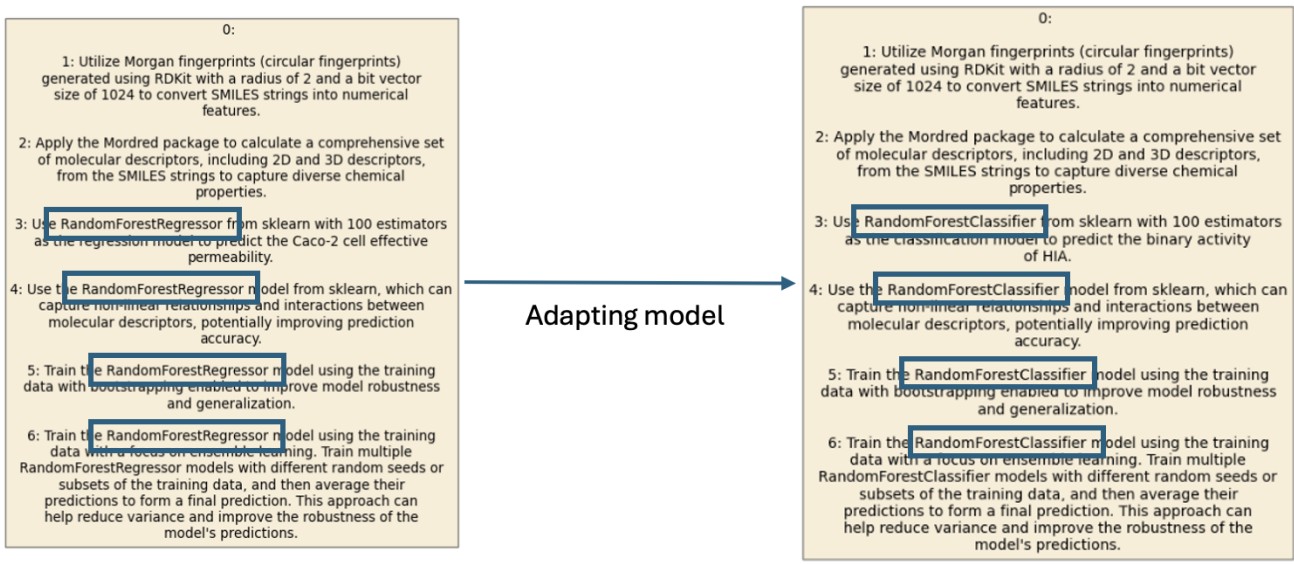

Figure 20: Example of the adapter LLM adapting the model from a regression task to a classification task.

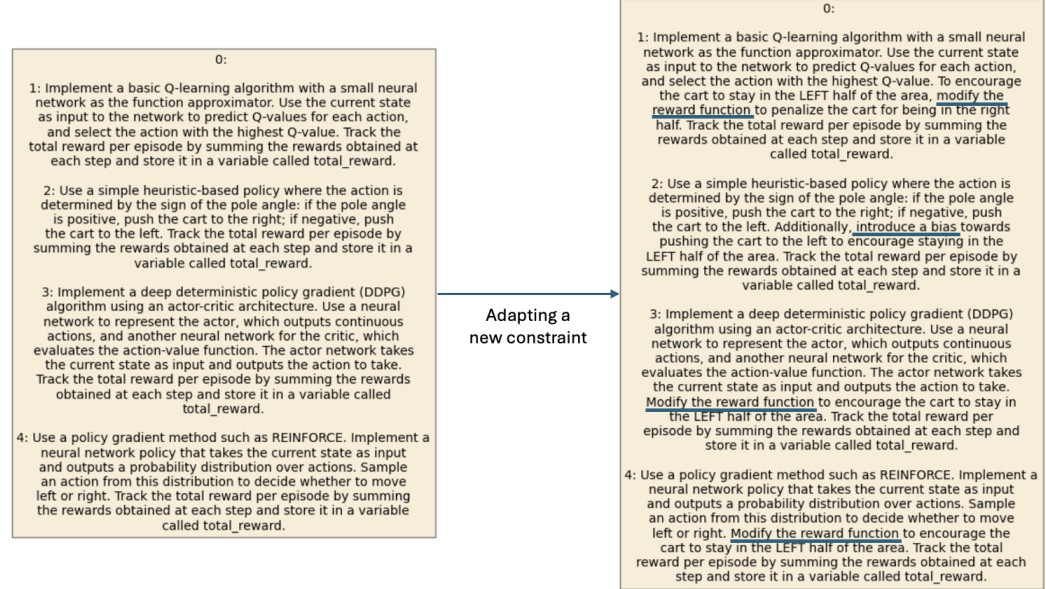

Figure 21: Example of the adapter LLM adapting to constraints of the new task in the case of cartpole balancing. From the original problem, the adapter modifies the methods to bias the cart to stay in the left of the arena.

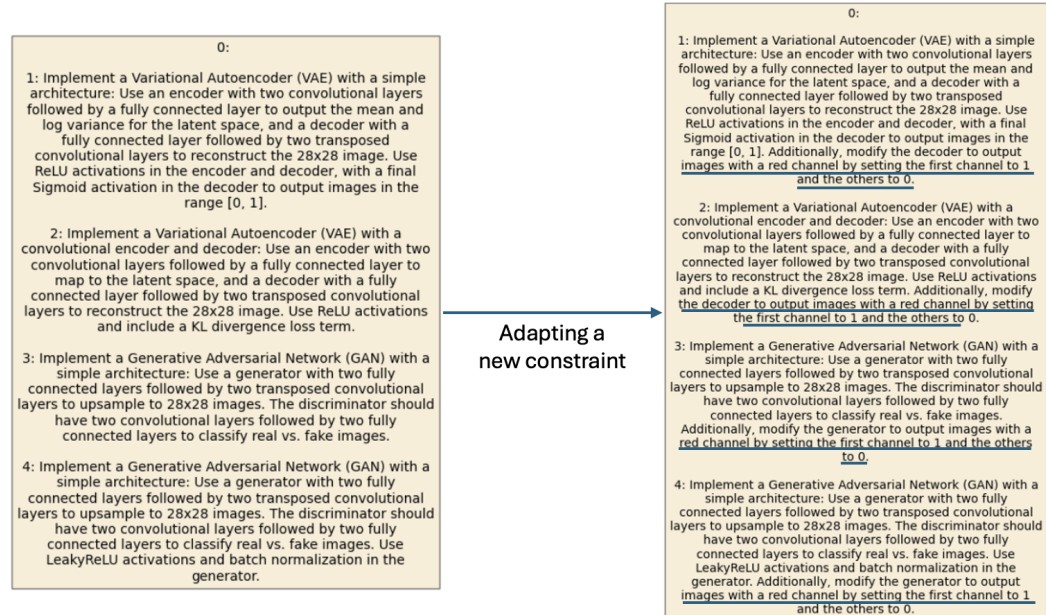

Figure 22: Example of the adapter LLM adapting to constraints of the new task in the case of MNIST image generation. From the original problem, the adapter modifies the decoder output with post-processing.

## M. Data interpreter prompts

We obtained these prompts from the Github repository of MetaGPT implementing the Data Interpreter (DeepWisdom, 2025). Note the specificity in the task types, with task-specific prompts such as image2web conversion (`IMAGE2WEBPAGE`). New task types would require similar manually constructed prompts. Also note the hand-crafted details such as model choices in `MODEL TRAIN PROMPT`. Despite the specific model choice instantiations, Data Interpreter mostly resolved to XGB and RF models, demonstrating the benefit of encoding options in a more explicit data structure like FoO. **Note the in-depth, specific details provided for each task type, such as how and which columns to process, or which models to use for specific tasks.** In contrast, the next section demonstrates our LLM prompts which are more general in comparison.

---

**Prompt: Data-Interpreter**

```
# Prompt for taking on "eda" tasks
EDA_PROMPT = """
The current task is about exploratory data analysis, please note the following:
- Distinguish column types with `select_dtypes` for tailored analysis and
    visualization, such as correlation.
- Remember to `import numpy as np` before using Numpy functions.
"""

# Prompt for taking on "data_preprocess" tasks
DATA_PREPROCESS_PROMPT = """
The current task is about data preprocessing, please note the following:
- Monitor data types per column, applying appropriate methods.
- Ensure operations are on existing dataset columns.
- Avoid writing processed data to files.
- **ATTENTION** Do NOT make any changes to the label column, such as
    standardization, etc.
- Prefer alternatives to one-hot encoding for categorical data.
- Only encode or scale necessary columns to allow for potential feature-specific
    engineering tasks (like time_extract, binning, extraction, etc.) later.
- Each step do data preprocessing to train, must do same for test separately at the
    same time.
- Always copy the DataFrame before processing it and use the copy to process.
"""

# Prompt for taking on "feature_engineering" tasks
FEATURE_ENGINEERING_PROMPT = """
The current task is about feature engineering. when performing it, please adhere to
    the following principles:
- Generate as diverse features as possible to improve the model's performance
    step-by-step.
- Use available feature engineering tools if they are potential impactful.
- Avoid creating redundant or excessively numerous features in one step.
- Exclude ID columns from feature generation and remove them.
- Each feature engineering operation performed on the train set must also applies
    to the dev/test separately at the same time.
- **ATTENTION** Do NOT use the label column to create features, except for cat
    encoding.
- Use the data from previous task result if exist, do not mock or reload data
    yourself.
- Always copy the DataFrame before processing it and use the copy to process.
"""

# Prompt for taking on "model_train" tasks
MODEL_TRAIN_PROMPT = """
The current task is about training a model, please ensure high performance:
- For tabular datasets - you have access to XGBoost, CatBoost, random forest,
    extremely randomized trees, k-nearest neighbors, linear regression, etc.
- For image datasets - you have access to Swin Transformer, ViT, ResNet,
    EfficientNet, etc.
- For text datasets - you have access to Electra, DeBERTa, GPT-2, BERT, etc.
```

```
- Avoid the use of SVM because of its high training time.
- Keep in mind that your user prioritizes results and is highly focused on model
    performance. So, when needed, feel free to use models of any complexity to
    improve effectiveness, such as XGBoost, CatBoost, etc.
- If non-numeric columns exist, perform label encode together with all steps.
- Use the data from previous task result directly, do not mock or reload data
    yourself.
- Set suitable hyperparameters for the model, make metrics as high as possible.
"""

# Prompt for taking on "model_evaluate" tasks
MODEL_EVALUATE_PROMPT = """
The current task is about evaluating a model, please note the following:
- Ensure that the evaluated data is same processed as the training data. If not,
    remember use object in 'Done Tasks' to transform the data.
- Use trained model from previous task result directly, do not mock or reload model
    yourself.
"""

# Prompt for taking on "image2webpage" tasks
IMAGE2WEBPAGE_PROMPT = """
The current task is about converting image into webpage code. please note the
    following:
- Single-Step Code Generation: Execute the entire code generation process in a
    single step, encompassing HTML, CSS, and JavaScript. Avoid fragmenting the code
    generation into multiple separate steps to maintain consistency and simplify
    the development workflow.
- Save webpages: Be sure to use the save method provided.
"""
```

## N. LLM prompts used in our framework

We present the prompts passed into the different LLM-based components of our framework. Note that our prompts are more general in comparison to frameworks like the Data Interpreter. **Importantly, we do not make any assumptions about task types like the Data Interpreter.**

---

**Prompt: Planner**

```
You are an expert ML scientist provided with the following task: {task}

Think through the task step-by-step and produce a highly detailed list of steps
involved in solving this task.

Do not include any code in your response.
```

---

**Prompt: Planner – Filter Mode**

```
You are an expert ML scientist provided with a task and a high-level plan for
solving the task.

Task: {task}
High-level plan: {plan}

Your goal is to rank the steps in the high-level plan based on their relevance to
accuracy on the task.

First, you must assess each step in the plan and identify whether it can impact
accuracy on the task.
```

Second, for all the steps that can impact accuracy on the task, you must rank the steps from most impactful to least impactful.

---

### Prompt: Adapter

You are provided with a task. You are also provided with a high-level plan for solving a different, but related task. Your goal is to adapt the steps in the plan for the new task.

Task: {task}
Plan: {plan}

For each step in the plan, you must identify if that step needs to be adapted for the current task.

If adaptation is necessary, modify the step to adapt it to the new task.
If no adaptation is necessary, leave the step as is.

Please do not add or remove any steps. Return the adapted plan.

---

### Prompt: Option Generator

You are an expert ML scientist provided with a task, and one step from a high-level plan for solving this task. You are also given implementation details for all of the steps prior to the current step.

task: {task}
prior steps: {s}
current step: {step}

Your goal is to generate diverse options for implementing the current step if it impacts final accuracy.

Firstly, you must assess whether the implementation of the current step can impact final accuracy on the task.

If it is expected to impact final accuracy, you must:

First, think as broadly as you can to generate {num_options} diverse and distinct options for implementing the current step, while taking into account the prior steps.

Second, you must verify that each of your generated options do not conflict with any of the prior steps.

Third, you must regenerate alternatives for any options that conflict with prior steps.

Please follow these instructions in generating your response:

1. You must maximize the diversity across all your generated options.
2. Your choices must be very specific so that a programmer can implement it. For e.g., instead of specifying "Use features", you must specify "Use features X, Y, Z."
3. Your choices should not modify any of the prior steps.
4. Do not repeat any of the prior steps.
5. Do not include code in your response.

---

**Prompt: Consistency Checker**

```
You are given a plan of steps for solving a task, and a new step. Your goal is to
verify if the information contained in the new step contradicts any of
previous steps of the plan.

plan: {plan}
new step: {step}

The following are examples of a contradiction:
1. If the new step references a different model than the previous steps in the plan.
2. If the new step references a different feature than the previous steps in the
plan.

First, assess whether the new step contradicts previous steps in the plan as
described above.

If the new step contradicts other steps in the plan, return the word: yes
If the new step does not contradict other steps in the plan, return the word: no
```

---

**Prompt: Code Generator**

```
You are an expert AI programmer provided with a task and a plan of steps for
accomplishing the task:

task: {task}
plan: {plan}

You are also given specific implementation details for some of the steps in the
plan: {walk}

Your goal is to stitch the plan together to create a complete Python code.

You must ensure to incorporate the specific implementation details into your final
code.

You must follow the template provided in the task description as closely as
possible and integrate the implementation details into the provided template.

You must not make any functional changes to the provided implementation details
themselves.

Your code must be complete and should not leave any additional steps for the user
nor raise any errors.
```

---

**Prompt: Code Generator – Reflective Debugging Mode**

```
You are provided with a code and an error message from running the code. Please
debug the code systematically to fix the error.

For additional context, you are also provided with the original user prompt
indicating the task that the code is trying to achieve.

Original prompt: {task},
Error message: {error},
Code: {code}

The error may also be syntactical in which case you must fix the syntax
appropriately.
```

# O. User inputs to our framework

We show examples of user inputs for the different tasks. We provide code template in our prompts (similar to DS-Agent (Guo et al., 2024)) to: a) **standardize evaluations between baselines** by removing differences in how the models get evaluated; b) to avoid issues with models not knowing what column names exist in the dataset, how to access them or other API issues. This can be alleviated by connecting the model to external tools that would allow it to inspect datasets, information contained therein, or to reason about APIs. Similarly, we pre-specify packages to use, such as Sklearn or Pytorch to prevent import errors when executing code (and to prevent autonomous installation of packages in our environment).

---

**User Input Example for TDC task**

```
Your task is to write Python code for the following task. The dataset name and task
description is provided below.

Dataset name: {task_name}
Task description: {description}

Please follow these instructions:
1. Follow the template in the example code shown below, and do not add any new print
statements to it.
2. Your code must be complete and executable without needing additional user
intervention.
3. Please use Sklearn or Pytorch packages only for your ML implementations.

Here is an example code snippet showing how to load and evaluate a dataset with the
name Caco2_Wang":

```
from tdc.benchmark_group import admet_group
group = admet_group(path = 'data/')
predictions_list = []

# For reproducibility
for seed in [1, 2, 3, 4, 5]:
    benchmark = group.get('Caco2_Wang')
    # all benchmark names in a benchmark group are stored in group.dataset_names
    predictions = {}
    name = benchmark['name']
    train_val, test = benchmark['train_val'], benchmark['test']
    train, valid = group.get_train_valid_split(benchmark = name, split_type =
    'default', seed = seed)
    # NOTE: For the dataset, column names are 'Drug' (for the input SMILES strings)
    and 'Y' (for the output labels)

    # ------------------------------------------- #
    #  Train your model using train, valid, test    #
    #  Save test prediction in y_pred_test variable #
    # ------------------------------------------- #

    predictions[name] = y_pred_test
    predictions_list.append(predictions)

results = group.evaluate_many(predictions_list)
print(results)
```
```

---

**User Input Example for RL task – optional constraint specification shown**

```
Your task is to solve a classic reinforcement learning problem where the goal is to
balance a pole upright on a cart that can move left or right.
```

```
# We add this for the variant of the problem
However, an important constraint in this problem is that your cart must try to stay
in LEFT half of the area as much as possible.

Follow the template below for your code:

```
import gym
import numpy as np
import torch
import random
from gym.wrappers.monitoring.video_recorder import VideoRecorder
import matplotlib.pyplot as plt

# For reproducibility
SEED = 42
random.seed(SEED)
torch.manual_seed(SEED)
np.random.seed(SEED)

# Create the environment
env = gym.make('CartPole-v1', render_mode="rgb_array")
video = VideoRecorder(env, "Cartpole-video.mp4")

# State space has four values -- [cart position, cart velocity, pole angle, pole
    angular velocity]
# Action space has two values -- {0: "push cart to the left", 1: "push cart to the
    right"}
for episode in range(500):
    state, _ = env.reset(seed=SEED)
    total_reward = 0.0
    # FILL OUT: Solution for the cartpole problem
    # FILL OUT: Predict action using model
    next_state, reward, done, _, _ = env.step(action)
    # FILL OUT: Track total reward per episode in a variable called total_reward

# Plot total reward vs. episode
plt.plot(total_reward)
plt.xlabel("Iteration")
plt.ylabel("Total Reward")
plt.savefig("total_reward.png")

# Print total_reward from the last episode
print("Final episode reward: ", total_reward)

state, _ = env.reset(seed=SEED)
done = False
while not done:
    # FILL OUT: Predict action using model
    state, _, done, _, _ = env.step(action)
    env.render()
    video.capture_frame()

video.close()
env.close()
```
```

**User Input Example for Clustering Task**

```
Your task is to perform clustering of gene expression data for a given dataset. You
are  provided the gene expression profiles for 2321 patients, on 14208 genes.
You must cluster them into 7 clusters.

Please visualize the generated clusters and save the visualizations.

Please follow the provided template to complete this task.
You must restrict yourself to Pytorch or Sklearn packages for this problem.
Please ensure that your code does not require any additional steps from the user.

Here's a template code to follow:

import pandas as pd
import numpy as np
from sklearn.metrics import silhouette_score

def compute_ssi(data, predicted_labels):
    print(f"Final Silhouette Score: {silhouette_score(data, predicted_labels)}")

def load_dataset():
    data = pd.read_table("Leukemia-clean.txt", sep = "\t")
    data["disease"] = np.where(data["disease"] == "Diabetes_Type_I" , "Diabetes",
    data["disease"])
    data["disease"] = np.where(data["disease"] == "Diabetes_Type_II" , "Diabetes",
    data["disease"])
    other = ['CML','clinically_isolated_syndrome', 'MDS',
    'DS_transient_myeloproliferative_disorder']
    data = data[~data.disease.isin(other)]
    df = data.drop("disease", axis=1)
    df = df.drop("GSM", axis=1)
    df = df.drop("FAB", axis=1)
    return df.to_numpy()

# There are 14208 columns (features) and 2321 samples
data = load_dataset()

# TO FILL --- Perform clustering ---- #
# TO FILL --- Predict labels for each sample and save into predicted_labels ---- #

compute_ssi(data, predicted_labels)

# TO FILL --- Save the clustering visualizations into a file called "clusters.png"
```

**User Input Example for English-French Translation Task**

```
We would like to build a language model that can translate from English to French
(en-fr).
Please use ROUGE score to evaluate the result.
Print the final ROUGE score as "Final Rouge Score: <print final rouge metric>".

1. You must load and use the opus-100 dataset (en-fr) as shown below.
2. You must restrict yourself to Pytorch or HuggingFace packages only.
3. Your code must be complete, and should not leave additional steps for the user.
4. Please restrict any training or fine-tuning to 1 epochs.
5. Please run all models and tokenzations on the GPU.

Example of loading the dataset:
from datasets import load_dataset
from torch.utils.data import Dataset, DataLoader
```

```
dataset = load_dataset("Helsinki-NLP/opus-100", "en-fr")

# Can be modified if needed.
class CustomDataset(Dataset):
    def __init__(self, dataset, transform=None):
        """
        Args:
            data (array-like): Data samples.
            targets (array-like): The corresponding labels for the data samples.
            transform (callable, optional): Optional transform to be applied on a
    sample.
        """
        self.data = dataset #[sample['translation']['en'] for sample in dataset]
        #self.targets = [sample['translation']['fr'] for sample in dataset]
        self.transform = transform

    def __len__(self):
        """Returns the total number of samples."""
        return len(self.data)

    def __getitem__(self, idx):
        """Generates one sample of data."""
        sample = self.data['translation'][idx]['en']
        target = self.data['translation'][idx]['fr']

        if self.transform:
            sample = self.transform(sample)

        return sample, target

train_dataset = CustomDataset(dataset['train'])
val_dataset = CustomDataset(dataset['validation'])
test_dataset = CustomDataset(dataset['test'])
```

**User Input Example for Traveling Salesman Problem**

```
You must solve a traveling salesman problem: Given a 2d matrix distance[][] of size
n (i.e.,  number of cities) where distance[i][j] denotes the distance from city i to
city j.

The task is to complete a tour from a city (0-based index) to all other cities such
that we  visit each city exactly once and then come back to starting city at minimum
total distance.
You should evaluate your approach on time taken to complete.

You can follow the template below for guidance:

import numpy as np
import torch
import random

# For fixing the TSP instance generated by generate_tsp() function
SEED = 42
random.seed(SEED)
torch.manual_seed(SEED)
np.random.seed(SEED)

def get_path_length(path, distances):
    # Path is a list of city numbers indicating the path to follow
```

```
    total_path_length = 0
    for i in range(len(path) - 1):
        start = path[i]
        end = path[i + 1]
        total_path_length += distances[start][end]
    return total_path_length

def generate_tsp(num_cities):
    """Generates a random TSP instance with given number of cities."""

    # Create random coordinates for cities
    coordinates = np.random.rand(num_cities, 2) * 100

    # Calculate distances between cities using Euclidean distance
    distances = np.zeros((num_cities, num_cities))
    for i in range(num_cities):
        for j in range(num_cities):
            if i != j:
                distances[i, j] = np.linalg.norm(coordinates[i] - coordinates[j])

    return distances

# Example usage:
num_cities = 10
distances = generate_tsp(num_cities)
optimal_path = None
time_taken = None  # Track time taken by the proposed approach

# TO FILL -- Predict optimal path, where optimal_path is a list of city numbers
    ranging from 0 to num_cities

assert len(optimal_path) == num_cities + 1, "Path does not visit all the cities"
assert len(set(optimal_path)) == num_cities, "Path does not visit all cities once"
print("Optimal Path: ", optimal_path)
print("FINAL Metric - Time taken: ", time_taken)
```

