# OpenReview forum: "Flow-of-Options: Diversified and Improved LLM Reasoning by Thinking Through Options"
_ICML.cc/2025/Conference — ICML 2025 poster_

### Official Review · Reviewer_cGuq · 2025-03-11

**Overall Recommendation:** 4

**Summary:**

The paper introduces the Flow-of-Options (FoO) framework, a structured reasoning approach for large language models (LLMs) that systematically generates and evaluates multiple decision options at each step. Instead of following a single reasoning path, FoO constructs a directed acyclic graph (DAG) where each node represents an option, and edges capture transitions between different decisions. The framework is applied to a range of tasks, including machine learning automation, therapeutic chemistry, reinforcement learning, and symbolic reasoning, aiming to improve decision-making by diversifying the exploration process. Additionally, FoO integrates case-based reasoning (CBR) to reuse past solutions for efficiency. The proposed approach is tested across multiple domains, comparing its performance against existing LLM-based agent systems and AutoML frameworks.

**Claims And Evidence:**

This paper has several claims about the performance of FoO framework, including performance, structured reasoning, generalization, and computational costs. The authors used a variety of tasks and other frameworks to compare, showing a robust advantage in performance and capacity in structured reasoning.

However, regarding computational costs, it may be oversimplistic to compare the LLM costs simply. Though the paper claims each task cost less than $1, however, implementing and running solutions in parallel can sometimes be very computationally intensive as well. Finetuning a language model is also expensive, but it may directly or quickly provide a good enough solution without testing all possible candidates. Therefore, it's hard to say which one is more computationally efficient.

The authors also mention such a framework has a stronger generalizability. However, this framework does not generate new insights. By asking LLM to list out options, this framework is more like implementing these options and validating their performance to choose the best one. This will largely depend on the capacity of the base LLM. If the LLM is not working creatively or has a really less representative or out-of-distribution task scenario, it would largely limit the performance of this framework in a more challenging, unsolved, and even unseen task. Though other frameworks may not overcome this difficulty as well, as an agent workflow, it should not only validate their actions, but also learn from a closed feedback loop to refine their policy, which is lacking in the current paper.

**Essential References Not Discussed:**

Some discussions about optimal algorithms in each task (not simply compare the existing agent framework), as well as human cognitive findings would be necessary.

For example, when talking about varying different tasks, what if adding reasoning models with tool use like o3-mini-high or deepseek-R1? For some specific tasks, what if using Bayesian Optimization in ML tasks, rather than asking the model to propose different options? There are some known other model pipelines as well as algorithms to achieve sota performance in relevant tasks. The agent framework may need to consider how this agent framework compares with those domain expertise solutions, but not only limited to agent frameworks.

On the other hand, discussion about how humans solve this task and potentially introducing a human baseline is also meaningful. What if humans can guide the model better by step-by-step feedback? What kind of heuristics do humans have that may be useful or harmful to the agent execution?

Example references:
Kahneman, D., & Tversky, A. (1979). Prospect theory: An analysis of decision under risk. Econometrica.
Evans, J. S. B. T., & Stanovich, K. E. (2013). Dual-process theories of higher cognition: Advancing the debate. Perspectives on Psychological Science.

**Experimental Designs Or Analyses:**

The experimental design and analyses are robust within a variety of tasks as well as model/agents comparison. However, the tasks can be more challenging (for new knowledge/solution discovery) to emphasize the value of this framework.

**Methods And Evaluation Criteria:**

The paper adopts a wide range of tasks, and the evaluation criteria are consistent with the task property. The workflow shows a generalizability of task domains. However, just I mentioned, if working on an even more challenging and creative task, this may limit the task performance.

**Other Comments Or Suggestions:**

I have no other comments or suggestions.

**Other Strengths And Weaknesses:**

Weakness: There are really a wide range of tasks, especially across domains. If in the main figures' captions, those short names could be specified, it would be much easier for readers to understand what the tasks mean.

**Questions For Authors:**

1. I noticed that in the supplementary information, there is one baseline about human coding on TDC leader board (Figure 14). I don;t quite get what the figure tries to reveal, are they saying this agent framework is better or worse than the human baseline?

2. I was wondering what if asking humans to guide the model to do similar tasks, for example, give LLMs compositional problems, ask LLMs to list out options and implement in parallel, what would it be better or worse than the current case? This question wants to ask how much the agent framework can work in the real practical problems compared with humans, with the same core LLM that implements codes or proposes options. Since we can always replace the core model with stronger ones, but we need to figure out how much role the workflow currently plays in the problems.

**Relation To Broader Scientific Literature:**

This paper proposes a FoO (flow-of-options) agent framework, optimizing multiple tasks of domain performance compared to the previous traditional frameworks. Though the agent workflow shows promising advantages in performance among these tasks, it does not essentially implement a learning and actively interacting agent, which is important for the stronger AI development. Current pipeline still primarily remains on using a known structure of knowledge to solve known structure of problems, which are more like picking up a best known solution, not coming up with an even better one. In this sense, the impact is limited.

**Theoretical Claims:**

The paper is mainly empirical. The limited theoretical claims come at the definition of FoF as well as the DAG nodes. I don't see any problems there.

---

> ### Author Rebuttal · Authors · 2025-03-29
>
> We thank the reviewer for their feedback. We have consolidated the reviewer concerns into the following topics and will include them in our updated paper.
>
> # 1. Computational cost compared to fine-tuning
> Fine-tuning introduces two additional implicit costs that do not impact our approach:
> - Fine-tuning requires a dataset that is reasonably large and diverse. Ensuring quality, quantity, and diversity of the dataset can present a high cost.
> - Fine-tuning also requires the model weights to be accessible which is not always the case. This enables a broader applicability of our approach.
>
> # 2. Tool use
> **Tool use is synergistic with Flow-of-Options**. In our paper, **FoO seeks to improve the base reasoning capabilities of the LLMs _even in the absence of tools_**. However, tools can be a *force multiplier* in our work. We conducted a small experiment to illustrate this further. We incorporate external research paper retrieval as a tool with FoO. The tool retrieves and provides two papers as context into the Option Generator LLM, for the example task of fine-tuning protein language models. We note the two retrieved papers incorporated into the option generator’s context: ProtBERT (Elnaggar et al. 2021) and ESM (Rives et al. 2021). Following is the resultant FoO produced: https://ibb.co/GQL0P06N. We see that the nodes 1 and 2 in the FoO incorporates the information in the papers when proposing options. In this way, tool use is synergistic with our work.
>
> # 3. Creativity and Discovery
> This is indeed an interesting point. We believe Flow-of-Options incorporates “combinational creativity” as described in Boden 1998. For instance, in the case of the Drug-target Interaction problem (DTI task from Table 4), our approach proposes the following ML model architecture option: Linear → Swish → dropout → Linear → GeLU → Linear (for regression). This is not an existing model per se. Although the individual components, such as the linear and activation layers exist, the specific combination of these layers can be considered novel and performs well on the task.
>
> The individual nodes can also be combinations of existing methods. For instance, in the Drug Combination problem (DC task from Table 4), existing human baseline on the leaderboard computes features of the drug molecules using packages such as RDKit, combined with a feed-forward neural network for prediction. In contrast, our approach proposes a novel ML pipeline that combines feature extraction using ChemBERTa (an existing language model for computing the embeddings of drug molecules), with feed-forward neural network for prediction. Although the individual components exist, this combination can be considered novel. In this sense, Flow-of-Options can support combinational creativity and discovery.
>
> # 4. Synergy with humans
>
> ## Q1
> Figure 14 compares the performance of the ML approaches produced by FoO vs. human designed baselines on the TDC leaderboard. The figure shows that our approach mostly achieves > 80% of an expert human’s performance. In other words, on average, it is comparable to (though not always better than) a human expert. In some cases, it outperforms the human baseline. For instance, in the drug combination (DC) task from Table 4, our approach outperforms the human baseline on the TDC leaderboard. For naive users however, FoO can significantly democratize ML.
>
> ## Q2
> Currently, the LLM proposes the options. However, these options could be informed through human guidance. Similar to our demonstration on tool use incorporating external references, the option generation can be conditioned on user inputs. In this way, the Flow-of-Options data structure can be a synergy of LLM and human knowledge leading to discovery of novel combinations.
>
> # 5. FoO with learning
> Our current implementation of Flow-of-Options seeks to improve the base reasoning capabilities of LLMs. However, it may be complementary to approaches that incorporate closed feedback learning. For instance, it may be possible to learn over walks produced by the Flow-of-Options data structure. Since each walk is associated with a corresponding metric, it can be treated as a supervised learning problem, or incorporated with reinforcement learning. This would make for an interesting future exploration.
>
> # 6. Domain expertise solutions
> We explore two such algorithms (neither use LLM-based agents): AutoGluon (Erickson et al. 2020) for data science tasks in Section 4.1, and DeepMol (Correia et al. 2024) for TDC ADME-Tox tasks in section 4.2. DeepMol is a specialized framework for ADME-Tox domain, and AutoGluon is a specialized framework for typical Data Science domain. Both DeepMol and AutoGluon optimize for ML models without using any LLM agents. AutoGluon has demonstrated improvements over Bayesian Optimization based AutoML such as Auto-Weka and other AutoML optimization frameworks. Hence we chose AutoGluon as the sota model. DeepMol is currently one of the sota AutoML methods for TDC tasks.

---

> > ### Comment · Reviewer_cGuq · 2025-04-03
> >
> > Thanks to the authors for addressing my concerns. Most of my concerns are well addressed. I will update my score to 4 to support the acceptance of the paper.

---

> > > ### Author Response · Authors · 2025-04-03
> > >
> > > We sincerely thank the reviewer for raising the score and for their insightful feedback that has helped improve our paper.

---

### Official Review · Reviewer_zYfD · 2025-03-11

**Overall Recommendation:** 3

**Summary:**

The paper introduces Flow-of-Options (FOO), an agentic system designed for auto-ML. The core contribution is a framework based on fully-connected network structures of step-by-step solution paths generated by LLMs.  The framework is evaluated comprehensively on multiple domains including standard data science tasks, therapeutic chemistry tasks, reinforcement learning, and image generation.

**Claims And Evidence:**

The paper presents several compelling results but some key claims lack sufficient supporting evidence:

* The claim about the framework's effectiveness would be strengthened by including more comprehensive ablation studies and hyperparameter analysis in experiments. For example, it is important to include comparisons across different LLM backbones, as well as sensitivity analysis of the hyperparameters for both the baseline methods and the proposed approach. Without these analyses, it's difficult to assess the robustness and generalizability of the performance improvements.
* Some claims in the paper would be better-supported by control studies, such as directly comparing the proposed fully-connected network structure against alternative architectures (trees, standard DAGs).
* Figures 2 and 7, which aim to use word clouds to show improved solution diversity, are weak and unconvincing. A more systematic quantitative analysis of solution diversity would be necessary to support this central claim of the paper.

**Essential References Not Discussed:**

NA

**Experimental Designs Or Analyses:**

* The evaluation is primarily conducted with GPT-4o as the underlying LLM. More comprehensive evaluation across different LLM architectures would be needed. (There are some very rough comparisons with GPT 3.5 in the supplemental, but it didn't compare with other baselines and lacks enough details).
* There are more ablation studies that can be done, such as evaluating the performance of the planner component and the consistency checker. It will let the readers have a better idea of which aspects of the framework are most critical to its success.
* Overall, I do not feel like I learned enough insights from the results beyond performance numbers. The paper should design more experiments that provide deeper understanding of the system.

**Methods And Evaluation Criteria:**

* The method is sound and presented in detail. The network structure design choices are well-justified in comparison to previous works.

* However, it is not clear if the paper properly tunes hyperparameters for the baseline methods, which raises concerns about the fairness of comparisons.

**Other Comments Or Suggestions:**

NA

**Other Strengths And Weaknesses:**

NA

**Questions For Authors:**

* Can you design experiments to show the effectiveness of the planner and consistency checker and their impact on the performance?
* How does the performance scale with the depth and width of the network?
* How do you deal with the scenario when there are multiple ways to decompose the problem?

**Relation To Broader Scientific Literature:**

The paper proposes a fully-connected network to build agentic systems for auto-ML. It improves upon existing approaches in several specific ways:

* Compared to SELA FoO offers greater expressivity by using a fully-connected network structure instead of a tree structure and replaces SELA's computationally expensive Monte Carlo Tree Search with a more efficient traversal mechanism.

* Compared with Data Interpreter, FoO provides a guarantee of acyclicity in its network structure since LLMs are only used to generate options, not to construct the network itself.

**Theoretical Claims:**

NA

---

> ### Author Rebuttal · Authors · 2025-03-29
>
> We thank the reviewer for their constructive feedback. We will incorporate the suggestions into our paper.
>
> # Q1
> We measure the reduction in execution time when consistency checker is added to identify invalid paths (as opposed to w/o it), and planner when the adapter is added (as opposed to w/o it). We also measure costs for each component when it is added. We measured these on average for tasks from Table 1. The results are noted here: https://ibb.co/7d4vmw7f. Adding these elements offer performance benefits at minor additional cost. We also performed a cost ablation for all components: https://ibb.co/LXgW5wBs
>
> # Q2
> Please note our scaling performance response to Reviewer utUi (Q1)
>
> # Q3
> This is a good question. We believe there are two cases in this scenario:
> - Case 1: A single task plan suffices, but has conceptually different option decompositions: For instance, a string denoting a drug molecule can be processed using the specialized RDKit package which computes chemical properties of the molecule such as molecular weight. Alternatively, we can convert the strings to vectors via NLP methods. These are conceptually different. However, the task plan constructed by **the planner is sufficiently high-level that conceptually distinct options are supported via the option generator LLM**. See the example FoO: https://ibb.co/GXXgj5p. Options 1 and 2 are feature processing, denoting the RDKit method and "NLP style" vector embedding respectively.
> - Case 2: A single task plan cannot denote the different decompositions: The experiments in our paper currently do not fall into this category. However, it is possible to envision a *nested* Flow-of-Options in our future work, with FoO data structure over task plans as well. In this case, the depth will be $n=1$ and width $k$ denotes the different task decompositions. Internally, each task plan node can be captured similar to our work in this paper. Hence, a nested version of FoO can be explored for such problems.
>
> # Expanded evaluation on LLM backbones
> We provide additional results here: https://ibb.co/NRMNTw4. Please see response to Reviewer 4BTz (W4) for more details. We will expand further details around this in our paper as well.
>
> # Comparison to the tree and DAG
> The current implementation of Data Interpreter does not appear to save the produced data structure.  SELA produces a text formatted tree shown in the excerpt below:
> ```
> [Node 0-2]
> Engineer features if necessary to improve model performance. Additionally, generate a correlation matrix heatmap to identify highly correlated features, which might be candidates for removal or transformation.
> ```
> We convert this to a visualization of the tree for a data processing task and demonstrate it alongside FoO for the same task. Tree: https://ibb.co/NdzGtFYW. FoO: https://ibb.co/mVqnMpk4. We note the qualitative option diversity and connectivity improvements of FoO here.
>
> # WordClouds
> We chose WordClouds as a compact representation of the frequency and diversity of the model choices made by our approach. We'd be happy to replace the wordclouds with bar charts (for proportion of options, i.e. frequency) which is more quantitative: https://ibb.co/3yVLzv0d (For Fig 2), https://ibb.co/tTS4Zwmz (Fig 5)
>
> # Hyperparams of baselines
> We have performed hyperparameter tuning on the key *number of iterations* parameters for SELA, DS-Agent, and Data Interpreter prior to running them (the other frameworks do not include hyperparameters apart from LLM backbone to use).
> - *SELA*: Increasing the number of iterations would lead to a significant explosion in terms of time (5 → 6 iterations increased time from ~21 mins → to ~57 mins on average), but we did not note a corresponding improvement in accuracy beyond 5 iterations. This could be related to the complexity of the problems in our experiments. More complex problems may require more iterations of SELA (albeit at a significant time cost).
> - *Data Interpreter*: Increasing the number of iterations increases the amount of time (to a much lesser degree than SELA), but like SELA, it did not impact accuracy beyond 5 iterations. It is also worth noting that the execution failures in Data Interpreter did not correlate to the number of iterations (possibly because failures are related to the presence of cycles in the LLM-built DAG, noted as one of the shortcomings of DI).
> - *DS-Agent*: The number of iterations increases the amount of time, and we also noted an improvement in accuracy however, that would stabilize at about iterations 4 to 5. We set this to 5 to be consistent in our cost/time comparisons across all methods.
> - *Our approach*: For our approach, the number of iterations increases the amount of time with improvement in accuracy. For consistency with all the baselines, we fix number of iterations to 5.
>
> We generally chose 5 iterations to maintain consistency across all our comparisons on time and cost assessments. We note relative advantages of the different methods in Appendix F.

---

> > ### Comment · Reviewer_zYfD · 2025-04-03
> >
> > Thank you for your comment. I have increased my score.

---

> > > ### Author Response · Authors · 2025-04-03
> > >
> > > We sincerely thank the reviewer for raising the score and for their insightful feedback that has helped improve our paper.

---

### Official Review · Reviewer_4BTz · 2025-03-12

**Overall Recommendation:** 3

**Summary:**

This paper proposes FoO (flow of options) approach to diverse the LLM's reasoning paths. An FoO-based agentic system is developed for solving traditional machine learning tasks including regression, classification, reinforcement learning, and image generation tasks. The authors show that their framework outperforms the existing methods by a large margin with lower cost.

**Claims And Evidence:**

Yes.

**Essential References Not Discussed:**

NA

**Experimental Designs Or Analyses:**

This paper evaluate their framework on many traditional machine learning tasks with other baseline methods. I have some questions as below:

1. For Table 1, it seems that zero-shot is better than DS-Agent, AutoGluon, and SELA. This is a bit confused and raises concerns that these frameworks may not suitable for the considered tasks, and make me question about the fair comparison among different approaches?
2. Fig 6 seems irrelevant as FoO-based approach by design should improve upon the iterations. However, other approaches are kinda independent among different trials (please correct me if I am wrong here).

**Methods And Evaluation Criteria:**

Yes.

**Other Comments Or Suggestions:**

NA

**Other Strengths And Weaknesses:**

**Strengths**:
1. This work use LLMs to generate diverse options to explore.
2. The propose approach can effectively explore the optimal path for a give task.
3. The authors explore the tasks beyond classification and regression, although just two cases.

**Weaknesses**:
1. The development part might be not efficient as the proposed methods explore the every path in the graph. When the width of the tree is larger, some bad options will also be visited many times.
2. The evaluation needs to be more rigorous and can be further improved. The comparison may not fair enough based on the current results.
3. More RL or other ML tasks should be explored to demonstrate the generalization of the proposed method.
4. Only GPT-4o is evaluated, how about other open-sourced models?

**Questions For Authors:**

In table 3, I am a bit confused by the cost of SELA, it seems that SELA takes the longest time but the cost is not the highest?

**Relation To Broader Scientific Literature:**

I believe the proposed methods can be related to broader applications other than traditional ML tasks.

**Theoretical Claims:**

There is no theoretical claims in this submission.

---

> ### Author Rebuttal · Authors · 2025-03-29
>
> We thank the reviewer for their constructive feedback. We will incorporate the suggestions into our updated paper.
>
> # Experimental Designs or Analyses
> ## Point 1
> The documentation for DS-Agent, AutoGluon, and SELA note that they are indeed suited to tabular tasks similar to the ones explored in our experiments. This is noted in their corresponding papers. Our dataset is not specifically curated by us, but is rather an existing dataset obtained from (Guo et al. 2024). We note the following potential reasons for the Table 1 observations:
> - *AutoGluon*: AutoGluon explores a fixed set of models which is well suited for some, but not all the data science tasks. Indeed some of the problems in the dataset, such as language-based tasks, are currently not supported by AutoGluon (which is a shortcoming of this method). Nevertheless, AutoGluon is a quite popular framework and is also an example of a non-LLM based AutoML framework. Hence we felt it would be useful to incorporate it as one of the baselines in our experiments.
> - *SELA and DS-Agent*: Both these frameworks incorporate *self-correction* over the LLM proposed methods. The LLM proposes a method and then repeatedly reflects on it to modify the method and improve the result. In “Large Language Models Cannot Self-Correct Reasoning Yet” by Huang et al. ICLR 2024, the authors note LLM performance can degrade after self-correction. Accuracy drops as the number of repeated self-correction calls increase (Table 3 of the cited paper). Both SELA and DS-Agent perform $>1$ self-correction calls. Besides zero-shot, Data Interpreter (DI) does not incorporate this form of self-correction, and tends to perform better among the baselines (although it has other shortcomings as noted in the paper). **LLM self-correction is not needed with Flow-of-Options, where the different options are already encapsulated and systematically explored via the FoO data structure. However, this does not mean that SELA and DS-Agent are not suited for the types of tasks in our experiments, but rather that there is room for improvement here, which we believe Flow-of-Options contributes.** It is possible that if self-correction were removed from DS-Agent and SELA their performances *could* improve. However, we do not change the baseline implementations available on Github for the purposes of comparison.
>
> ## Point 2
> This is correct with the exception that DS-Agent (alongside FoO) also incorporates mechanisms to improve upon past iterations. The other approaches do not have this design element. Our intention with Figure 6 is to:
> - Add experimental and *quantitative* evidence to our claims of being able to improve over subsequent iterations.
> - Demonstrate that some of the baseline approaches fail in certain iterations (e.g., DeepMol and data interpreter).
>
> We wanted Figure 6 to demonstrate the benefits of our approach at the agentic design level in comparison to the baselines.
>
> # W1
> We seek to mitigate this with the beam search (Section 2: beam traversal, page 2), which selects the top $b$ options for exploration, thereby limiting the explored paths to the more promising options. **This prevents the bad options from being revisited**. We note additional computational improvements in Section 3.2, Page 5 (Also note response to Reviewer utUi Q1).
>
> # W2
> We hope that our response to Point 1 above regarding baselines in Table 1 offers additional context around our compared baselines. Please also note additional experiments reported for W4 to strengthen our evaluation.
>
> # W3
> Currently, in addition to the classification/regression tasks, the RL and image generation tasks, we explored the following tasks in Appendix B.2 and B.3 to further demonstrate generalizability:
> - Clustering
> - Machine Translation
> - Traveling Salesman task
> - Case study on a math problem
>
> We hope that these additional tasks help demonstrate broader generalizability of our framework.
>
> # W4
> We provide additional results for LLMs (including an open-source LLM) over a subset of the TDC tasks from Table 2: https://ibb.co/NRMNTw4. Arrows indicate whether lower or higher metrics are preferred. Our approach helps consistently improve performance across the newly added LLMs and also outperforms the baselines for the tasks. Note that for some cases "--" indicates that the model failed to produce working code within three attempts. Hence, we see that **in addition to improving the overall task performance compared to baselines, FoO can also help mitigate the failure rates in code generation**.
>
> # Q. SELA cost
> A lot of the time consumed in SELA is for the execution of the code from the MCTS rollouts. Each code execution is performed sequentially (it does not support parallelization like our framework -- parallelization is discussed in Section 3.2, Page 5) and therefore takes a significant amount of time. However, the code execution does not involve LLMs per se, therefore it does not reflect in the cost, which is associated with querying LLMs.

---

### Official Review · Reviewer_utUi · 2025-03-16

**Overall Recommendation:** 3

**Summary:**

This paper proposes Flow-of-Options, a planning method for LLM agents, that can effectively track an optimal path over the combinations of possible options. More formally, Flow-of-Options can be represented as a directed-acyclic graph (DAG) of depth n, where a node is an option and an edge is a path between options in a sequence. Flow-of-Options finds an optimal path by evaluating possible paths and updating values (the return of the path) in edges. This paper evaluates Flow-of-Options on 16 Data Science (DS) tasks and 17 Therapeutic Data Commons (TDC) tasks. Experiment results show that Flow-of-Options can achieve better scores than SELA (utilizing a MCTS-based planner) and Data Interpreter (utilizing a DAG-based planner) in DS tasks.

**Claims And Evidence:**

This paper proposes Flow-of-Options, a planning method for LLM agents. And, it provides comprehensive evaluation results on 16 Data Science (DS) tasks and 17 Therapeutic Data Commons (TDC) tasks. Also, the evaluation results support that Flow-of-Options can achieve better scores than other baselines such as DS-Agent, AutoGluon, SELA, Data Interpreter, and AutoGen on DS tasks.

**Essential References Not Discussed:**

This paper properly discusses related works.

**Experimental Designs Or Analyses:**

This paper provides comprehensive experiment results on 16 Data Science (DS) tasks and 17 Therapeutic Data Commons (TDC) tasks. Also, it provide the details on experiments in Section B in the Appendix. It seems that the experiment design is sound and valid. However, it seems that the experiments mainly deal with data science and chemical reaction tasks.

**Methods And Evaluation Criteria:**

### Strengths of Methods:

S1. Flow-of-Options has a structure that can effectively tack an optimal path over possible combinations.

### Weaknesses of Methods:

W1. Flow-of-Options seems to be overly customized for data science tasks which goal to find an optimal path consisting of feature engineering and model selection. I am not sure if Flow-of-Options generally works well over diverse tasks that is more complex.

W2. I am not sure what is a key advantage of Flow-of-Options over the exhaustive search over all combinations.

**Other Comments Or Suggestions:**

### Other Comments:
N/A

**Other Strengths And Weaknesses:**

### Other Strengths:
N/A

### Other Weaknesses:
N/A

**Questions For Authors:**

### Questions:

Q1. Can you provide the computational complexity of finding an optimal solution in Flow-of-Options? If the number of options (k) is large and the number of steps (n) is long, the computational complexity can be high, since the number of walks increases exponentially (k^n). Can you provide the average number of options in some example tasks? How do you control the number of options in each step?

Q2. In the Development phase of the FoO-based agent framework, how long does it take for the values in edges to converge?

Q3. Can you provide some details on beam search over Flow-of-Options?

**Relation To Broader Scientific Literature:**

This paper introduces Flow-of-Options, a DAG-based planning method for effective LLM agents. Designing an effective planning algorithm for LLM agents is one of important research areas, since LLM agents are widely applied into diverse domains.

**Theoretical Claims:**

This paper mainly proposes a method for effective reasoning of LLMs. It does not provide any theoretical claims.

---

> ### Author Rebuttal · Authors · 2025-03-29
>
> We thank the reviewer for their constructive feedback and will incorporate these suggestions into our updated paper.
>
> # W1
> While our paper is on *Application-Driven Machine Learning*, **FoO does not explicitly specify steps such as model selection or feature engineering, but rather adapts to the task plan that is produced**. We show this generalizability (beyond classification/regression of sections 4.1 and 4.2) as follows:
> - RL and Image Generation (Section 4.3) - uses a different ML pipeline than tasks in 4.1 and 4.2 (e.g., reward formulation)
> - Unsupervised clustering (Appendix B.2.1) – is a different model than the data science tasks
> - Machine Translation (Appendix B.2.2) – does not involve a specific feature engineering like the data science tasks (only tokenization)
> - Traveling Salesman Problem (Appendix B.2.3) – involves neither ML model selection nor feature engineering
> - Case study on a math problem for solving the complex solutions to $x^2 + 2x = i$ using FoO (Appendix B.3) - does not involve an ML model selection nor any feature engineering.
>
> In particular, the TSP and math problems are larger deviations from the typical data science pipeline.
>
> # W2
> **FoO enforces diversity in LLM solutions through compressed, interpretable representations that support memory of past explorations when combined with case-based reasoning**. Specifically, FoO offers the following advantages over exhaustive combinatorial search:
> - FoO acts as “memory” of key info on previously explored task solutions - can be saved, reused and adapted from one task to another via deployment (Section 3.2). Reusing FoO from task $T_1$ to $T_2$ is fast and achieves good results (Tables 1, 2 and 3). Hence, prior knowledge encapsulated in the FoO can be effectively reused for $T_2$, whereas an exhaustive search would have to be repeated for each task.
> - Exhaustive combinatorial search w/o FoO assumes expert knowledge to set up the combinations themselves. An expert ML scientist can enumerate $k$ options for each step in the task, but a naive user may lack this knowledge. FoO encapsulates the knowledge of LLMs enabling even naive users to specify their problem in natural language, without demanding expertise. Even for experts, FoO can serve as a "force multiplier”, potentially enumerating options that the expert may not have thought of.
> - The formulation of FoO also supports integration with tool use (Please see Reviewer cGuq Point 2).
>
> # Q1
> Average number of options $k = 4$ and average number of steps $n = 3$. These are specified as hyperparameters into the framework. The computational complexity of FoO is indeed dependent on $n$ and $k$ and we have currently implemented the following solutions to mitigate the computational complexity (From Section 3.2, Page 5):
> - *Parallelization*: We parallelize the executions of walks through the FoO so that even if there are a large number of walks, the computational time taken is reduced.
> - *Pruning*: We prune some of the low-scoring edges of the FoO (so that they are not explored) which also helps reduce the computational complexity.
>
> Lastly, the consistency checker (Section 2, page 3) identifies invalid paths in the FoO. Although the total number of paths are $k^n$, not all paths are valid (as $n$ increases, the proportion of invalid paths is also higher). Example of invalid paths are shown in Fig 4 (Page 3). Consistency checker empirically results in $\approx 22.8$% reduction in the number of paths to explore (Empirical results noted in response to Reviewer zYfD Q1). Hence, in practice, computational complexity scales quite differently. Please see measured time performance with $n$ and $k$ (averaged across three runs on the CW TDC task of Table 2): https://ibb.co/0VvqpBR5. From $n=1, k=3$ to $n=3, k=3$, the number of paths is $9\times$ more. However, corresponding time only scales by $\approx 7\times$ (this includes parallelization and consistency checking, but excludes pruning which can further improve the efficiency).
>
> # Q2
> The development phase took about 13.29 mins on average in our work for the data science tasks. The development time is dependent on the complexity of the task based on $n$ and $k$ as noted in Q1.
>
> # Q3
> In beam search, our goal is to narrow the search to the most effective set of options, by selecting the top $b$ options at each level, and exploring paths between them to discover potentially improved combinations of the top options. This can be visualized as exploration over a reduced FoO where the nodes are just the top performing options found thus far.
>
> In our experiments, we start with a full beam width of 100% (exploring all the options), and reduce it to the top 50% of the options in the last two iterations of development. In Appendix C (Figure 13), we demonstrate that beam search can discover new combinations of top performing options, resulting in improvements in the final performance of the methods.

---

### Decision · Program_Chairs · 2025-05-01

**Decision:**

Accept (poster)

**Comment:**

This paper introduces Flow-of-Options, a planning approach aimed at enhancing LLM reasoning for auto-ML applications.
The system works by representing plans as DAGs, iteratively evaluating potential paths, and updating values along each edge to determine the best sequence of options. Experiments across 16 Data Science and 17 Therapeutic Data Commons tasks show that Flow-of-Options outperforms traditional planners like SELA (based on a MCTS planner) and Data Interpreter (which also utilizes a DAG-planner).

Reviewers appreciated the method's soundness and detailed presentation, found the main design choices well-motivated and justified, and praised the comprehensive experimental validation. Initial concerns about the narrow customization of the method to narrow data science tasks were addressed by the authors, who provided satisfactory clarifications about the generalizability of the method.
Other inquiries by the reviewers including on computational cost, scalability, and relation to other agentic paradigms like tool use were also satisfactorily addressed by the rebuttals, following which all reviewers indicated scores in the accept range.